# Understanding Transfer Learning of RNA Foundation Models on Downstream Tasks

**Yuan Li** [1 2]   **Heng Yang** [2 3]   **Renzhi Chen** [4]   **Ke Li** [1 2 *]

## Abstract

Foundation models (FMs) pretrained on large-scale sequence data have emerged as a promising paradigm for RNA biology, yet the mechanisms underlying their transferability remain unclear. In this work, we conduct a systematic investigation of transfer learning in RNA FMs across structural and functional tasks. Our results demonstrate that frozen representations from pretrained RNA FMs are not universally transferable, and that the hierarchical feature reuse paradigm prevalent in computer vision does not generally extend to RNA FMs. Instead, pretraining primarily benefits downstream tasks by providing a favorable optimization initialization when pretraining and downstream objectives are well aligned, which accelerates convergence toward flatter minima associated with improved generalization. Overall, our findings characterize pretraining as an optimization prior whose effectiveness is governed by task alignment and model capacity, offering principled guidance for future RNA FMs.

## 1. Introduction

From the discovery of DNA to the large-scale sequencing of life forms, the faithful and rule-governed flow of biological sequence information from DNA to RNA and protein has constituted the central tenet of life science. (He et al., 2025; Crick, 1970). Beyond mediating the flow of genetic information from DNA to proteins, RNA exhibits functional diversity and participates in a broad spectrum of cellular processes, including metabolism, transport, signaling, and regulation. (Zou et al., 2024; Cech, 2000; Morris

& Mattick, 2014). RNA complexity stems from its intricate structural organization, encompassing base-pairing patterns and higher-order folding in two and three dimensions. Beyond structure, RNA displays diverse functional roles, from catalysis to gene regulation (Arora et al., 2025; Wan et al., 2011). Consequently, understanding how diverse RNA structures and functions emerge from a simple four-letter chemical alphabet is essential for elucidating complex cellular processes and the molecular mechanisms underlying disease (Ren et al., 2024; Esteller, 2011).

Inspired by the success of machine learning across diverse domains, substantial research efforts in recent years have focused on applying machine learning approaches to RNA-related tasks, spanning from traditional methods to deep neural networks (Linder et al., 2025; Karollus et al., 2021; Li et al., 2018b; Bogard et al., 2019; Fu et al., 2022; Gong et al., 2021). However, most existing approaches rely heavily on labeled data. As a result, models must be deliberately designed and tuned for specific tasks, making them difficult to transfer to other related applications. Due to the dynamic nature of RNA structures, only a limited number of experimentally resolved RNA structures are available in the Protein Data Bank (Berman et al., 2000). Moreover, task-specific functional annotations for RNA are often scarce(Townshend et al., 2021; Yao et al., 2019).

Although RNA structural and functional data are scarce, the rapid development of next-generation sequencing technologies has produced vast amounts of RNA sequence data (Satam et al., 2023; Ozsolak & Milos, 2011). Similar trends have been observed in natural language processing (NLP) and protein science, where foundation models (FMs) have demonstrated transformative impact (Dalla-Torre et al., 2025; Boshar et al., 2025). Representative foundation models in NLP include language models such as BERT (Devlin et al., 2019) and GPT (Radford et al., 2018), which learn general-purpose representations from large-scale unlabeled data and can be adapted to a wide range of downstream tasks. Protein FMs, when transferred to downstream tasks, have demonstrated competitive and often superior performance compared to previous approaches in tasks such as protein structure and function prediction, even under data-scarce conditions (Elnaggar et al., 2021; Brandes et al., 2022; Lin et al., 2023; Elnaggar et al., 2023).

[1]Living Systems Institute, University of Exeter, Exeter, EX4 4QD, United Kingdom. [2]Department of Computer Science, University of Exeter, Exeter, EX4 4RN, United Kingdom. [3]Hithink Research, Hangzhou, China. [4]Qiyuan Lab, Beijing, China. Correspondence to: Ke Li <k.li@exeter.ac.uk>.

*Proceedings of the 43$^{rd}$ International Conference on Machine Learning*, Seoul, South Korea. PMLR 306, 2026. Copyright 2026 by the author(s).

Motivated by the success of FMs in NLP and protein science, recent efforts have explored RNA foundation models as a promising approach for learning representations of RNA sequences (Yin et al., 2025; Chu et al., 2024; Celaj et al., 2023; Fradkin et al., 2025; Zhu et al., 2025; Shen et al., 2024; Yuan et al., 2024; Zhang et al., 2024b; Wang et al., 2025; Yang & Li, 2024). However, despite the widespread adoption of FMs in RNA downstream tasks, it remains unclear whether and why pretraining consistently improves transfer performance. Without a clear understanding of these mechanisms, the field risks blindly scaling models under assumptions of feature reuse that may not hold for biological sequences. Indeed, prior studies on protein foundation models have shown that current pretraining strategies do not scale uniformly across diverse aspects of protein biology (Li et al., 2024). To address this gap, we present the first systematic dissection of transfer learning mechanisms in RNA foundation models.

In this paper, our main contributions are:

- We present, to the best of our knowledge, the most comprehensive evaluation of transfer learning for transformer-based RNA FMs on a diverse suite of downstream tasks, covering both structure prediction and function prediction.

- Our experiments show that the usefulness of frozen last-layer representations in RNA foundation models is task-dependent. While frozen features outperform baselines on structure prediction tasks, they often underperform a simple one-hot encoding on function prediction tasks. This pattern suggests a misalignment in which MLM pretraining captures structural regularities but suppresses local sequence statistics critical for certain functional tasks.

- By tracking the $\ell_2$ distance of hidden layers from their initial states, we identify three distinct patterns of knowledge transfer. Unlike in computer vision, where shallow layers often encode general-purpose features that are preserved during transfer, our results suggest that stable low-level feature reuse is not universal.

- We show that when downstream tasks require substantial global adaptation, pretraining can act less as a source of preserved features than as an effective initialization. It steers fine-tuning toward flatter minima, yielding faster convergence and better generalization.

## 2. Related Work.

### 2.1. Pretrained RNA Foundation Models

Early efforts such as RNABERT (Akiyama & Sakakibara, 2022) and RNA-FM (Chen et al., 2022) showed that transformer-based pretraining can produce informative RNA

representations, including signals related to structure, function, and evolutionary information. Since then, transformer-based architectures have continued to dominate RNA foundation model research, most of which are encoder-only transformers pretrained using the MLM objective. Unlike the DNA domain or protein domain, RNA sequences are very scarce and scattered across various biological databases. Due to the scaling-up law, existing RNA foundation models are relatively smaller compared to protein language models (up to 100B parameters) (Zou et al., 2024; Lin et al., 2023; Hayes, 2024) . Even though, these works constantly predicted that RNA foundation models can learn the general biological information, evidenced by their better performance on downstream structural-related or functional-related tasks than the SOTA, after a few epochs of finetuning or even zero-shot setting. However, none of previous works have given system investigation on whether or why pre-training contributes to performance improvements during downstream finetuning. While some interpretability studies provide insights, they typically focus on the static properties of pretrained embeddings rather than the dynamic processes that govern knowledge transfer during fine-tuning. In this work, we narrow down our research of transfer learning mechanism of small- to medium-size transformer-based RNA foundation models.

### 2.2. Transfer Learning

While the transfer learning mechanisms of RNA foundation models have never been discussed, many related research have been done in computer vision, especially in transfer learning for medical imaging (Huix et al., 2024; Islam et al., 2022; Suganyadevi et al., 2022; Salehi et al., 2023; Usman et al., 2022). This is where we take inspiration from. For example, Neyshabur (Neyshabur et al., 2020) used centered kernel alignment (CKA) to measure the similarity between two output features of the same network that are trained separately from pretrained weights and initialized weights. They emphasized the role of feature reuse, and especially stated that modules in the lower layers are in charge of general features (Grigg et al., 2021; Yosinski et al., 2014). Raghu (Raghu et al., 2019) also look at per-layer representational similarity before and after finetuning. They showed that larger models change much less during training especially in the lowest layers, which emphasized that feature reuse was restricted to the lowest couple of layers. Matsokas (Matsoukas et al., 2022) examined the importance of feature reuse using both CKA and layer-wise $\ell_2$ distance analysis, which again provided strong evidence of feature reuse. The above works mainly centered computer vision domain, where convolutional neural networks and vision transfomers play main roles.

Related research have also been done in protein language models and genomic language models. Li (Li et al., 2024)

observed that almost all downstream tasks do benefit from pretrained models, while they uncoupled improvements in downstream performance from scaling properties, which told us blindly enlarge protein language model did not guarantee performance improvement. However, another recent work challenged traditional transfer learning expectations: Vishniakov (Vishniakov et al., 2024) found that pretrained genomic foundation models demonstrated limited or even no advantage over their random initialized counterparts across a wide array of downstream genomic tasks.

## 3. Datasets and Pretrained Models

We conducted systematic experiments on a diverse suite of downstream tasks using RNA foundation models of different sizes and pretraining datasets. The downstream tasks are summarized in Table 1, with additional details provided in Appendix Table 2. More experimental details can be referred to Appendix A.

### 3.1. Downstream Tasks

We evaluated a diverse set of tasks spanning both structure and function prediction, different types of distribution shift relevant to RNA tasks, and variations at both global and local sequence levels.

**Structure Prediction.** We consider two types of structure prediction tasks. The first is a token-level three-class secondary structure prediction (SSP) task, which includes curated datasets from OmniGenBench (Yang et al., 2025b) and BEACON (Ren et al., 2024), as well as the **Archive2** and **bpRNA** datasets (Danaee et al., 2018). The objective is to predict whether each nucleotide is paired or unpaired. The second type is structural score imputation (**SSI**), which aims to predict missing structural information within RNA molecules. Accurate structural score imputation provides more comprehensive structural information and is crucial for the development of RNA-based therapeutics and diagnostics. This dataset is obtained from the BEACON benchmark (Gong et al., 2021; Ren et al., 2024).

**Function Prediction.** We consider several RNA function prediction tasks drawn from the BEACON benchmark and RNAGym. Specifically, we use the non-coding RNA function classification (**ncRNA**) and mean ribosome loading (**MRL**) datasets from BEACON. In addition, RNAGym provides a collection of 70 deep mutational scanning assays spanning diverse RNA systems, including tRNAs, aptamers, ribozymes, and both coding and splicing-disrupting mRNAs. In this study, we select two representative fitness prediction datasets (Soo et al., 2021; Ding et al., 2024) from RNAGym, referred to as Ribozyme and mRNA for simplicity.

The ncRNA task aims to classify non-coding RNA molecules into functional categories such as microRNAs (miRNAs), long non-coding RNAs (lncRNAs), and small interfering RNAs (siRNAs). This task is formulated as a 13-class sequence-level classification problem, with accuracy used as the evaluation metric (Amin et al., 2019; Fiannaca et al., 2017). The MRL task predicts the mean ribosome loading value of a given RNA sequence, which reflects the level of mRNA translation into proteins (Sample et al., 2019; Leppek et al., 2018). This task is treated as a sequence-level regression problem and is evaluated using $R^2$.

In contrast, the Ribozyme and mRNA datasets are designed to measure the effects of local sequence variations. For each dataset, we construct two train–test splits that introduce different types of distribution shift, namely an in-distribution split and an out-of-distribution split.

For the Ribozyme dataset, we evaluate the following splits: **Ribozyme:** Sequences are randomly partitioned into $80\%$ for training and $20\%$ for testing. **Ribozyme-ood:** Models are trained on variants containing fewer than three mutations and evaluated on variants with a larger number of mutations.

Similarly, for the mRNA dataset, we consider the following splits: **mRNA:** Sequences are randomly partitioned into $80\%$ training and $20\%$ testing. **mRNA-ood:** Models are trained on variants with fewer than six mutations and tested on variants with more mutations.

### 3.2. Transfer Learning with RNA Foundation Models

While a number of RNA foundation models have been proposed, we focused on small to medium sized, encoder-only transformer-based models int this study. These models are trained using the popular BERT MLM task (Vaswani et al., 2017). During pretraining, a subset of tokens (15%) is randomly sampled for corruption. Among these selected tokens, the majority are replaced with a `[MASK]` symbol, while smaller fractions are substituted with random tokens or left unchanged. The model is then optimized to recover the original token identities at the corrupted positions.

RNA models primarily focus on capturing the structural and functional properties of non-coding RNA (ncRNA) and messenger RNA (mRNA). Early models like RNA-FM, RNA-MSM (Zhang et al., 2024a), and RNA-BERT (Akiyama & Sakakibara, 2022) utilize Single Nucleotide Tokenization (SNT) and are pre-trained on diverse datasets such as RNAcentral (rna, 2019) and Rfam (Griffiths-Jones et al., 2003). These models excel at capturing evolutionary information and structural motifs within non-coding sequences across multiple species. To address the complexity of protein-coding sequences, models like mRNA-FM (239M parameters) (Chen et al., 2022) and SpliceBERT (Chen et al., 2023) were developed. Specifically, 3UTR-BERT (Yang et al., 2024) focuses on the 3' Untranslated Regions (UTRs), which are critical for post-transcriptional

*Table 1.* A simplified overview of RNA tasks and datasets.

| Task Category | RNA Task | Metric | Task Type | Description |
|---|---|---|---|---|
| Structure Prediction | Archive2, bpRNA | ACC | Token Cls | Secondary structure prediction |
| | SSI | Pearson $R^2$ | Token Reg | Structural score imputation |
| Function Prediction | MRL | $R^2$ | Sequence Reg | Mean ribozyme loading prediction |
| | ncRNA | ACC | Sequence Cls | Non-coding RNA classification |
| | mRNA, mRNA-ood | MSE | Sequence Reg | Mutant fitness prediction |
| | Ribozyme, Ribozyme-ood | MSE | Sequence Reg | |

regulation. These models enable precise transfer learning for tasks such as splice site prediction and mRNA stability analysis. The PlantRNA-FM (Yu et al., 2024) and the OmniGenome (Yang et al., 2025a) series were pre-trained on the OneKP dataset (Matasci et al., 2014), with pretraining objectives that explicitly incorporate RNA secondary structure alignment. More details about the foundation models are in the Appendix Table 3.

## 4. Experimental Setup

### 4.1. Baselines and ablations

We consider two transfer learning strategies in this work: (i) training task-specific heads on representations extracted from the last hidden layer of pretrained models, and (ii) full fine-tuning of all model parameters. Throughout the paper, we use the following abbreviations to denote the four training settings: **Onehot**, training models from one-hot sequence embeddings; **RI-T**, training models from randomly initialized weights; **Frozen**, training task heads on frozen last-layer representations from pretrained models; **PT-FT**, fine-tuning all parameters on the downstream tasks starting from pretrained weights.

**Onehot.** To determine the performance achievable without deep feature extraction, we evaluate a linear classifier trained directly on one-hot encoded sequences. This baseline serves to identify tasks that rely solely on trivial sequence statistics, establishing a lower bound that helps to isolate and measure the unique contribution of the deep models' ability to capture complex, long-range dependencies.

**Frozen.** To evaluate whether pre-training provides off-the-shelf features that align well with downstream tasks, we used linear probing analysis to test performance of directly using frozen last hidden representations of RNA FMs.

**RI-T.** To test whether the performance gain comes from the benefit of better initialization, we trained each model from scratch using randomly initialized weights. This also isolates the impact of the model architecture—such as the attention mechanism's ability to model long-range dependencies—from the knowledge acquired during pretraining.

**PT-FT.** To ensure a fair comparison across the diverse model sizes, we maintain a consistent training protocol. For all tasks, we attach a minimal task-specific head (a single linear layer followed by activation) to the backbone.

All experiments in this study are conducted based on the OmniGenBench platform, a unified benchmark and evaluation package for genomic foundation models proposed by Yang (Yang et al., 2025b). OmniGenBench provides standardized datasets, training pipelines, and evaluation protocols across a wide range of genomic and transcriptomic tasks, enabling fair and reproducible comparisons between different modeling and transfer learning strategies. The training details can be found in the Appendix A.

### 4.2. Analysis Metrics

#### 4.2.1. BLOCK-WISE $\ell_2$ DISTANCE.

One approach to studying feature reuse is to quantify how model parameters deviate from their pretrained values during fine-tuning. While prior work typically measures overall parameter displacement by comparing only the initial and final weight configurations (Matsoukas et al., 2022), such analyses overlook the temporal dynamics of adaptation. In contrast, our methodology captures the full learning trajectory throughout fine-tuning. Specifically, we record the $\ell_2$ distance between the weights of each transformer block and their corresponding initial values at every training epoch. This fine-grained analysis enables us to characterize not only the extent of parameter change in each layer, but also the rate at which adaptation occurs. A small $\ell_2$ distance indicates substantial feature reuse with minimal task-specific modification, whereas a rapidly increasing $\ell_2$ distance reflects fast adaptation and pronounced specialization for the downstream task.

#### 4.2.2. LOSS LANDSCAPE GEOMETRY.

To qualitatively characterize the solutions obtained from different initialization strategies, we visualize the geometry of the loss surface in the neighborhood of a converged model parameter vector, denoted as $\theta^*$. Following the approach of Li et al. (Li et al., 2018a), we project the high-dimensional

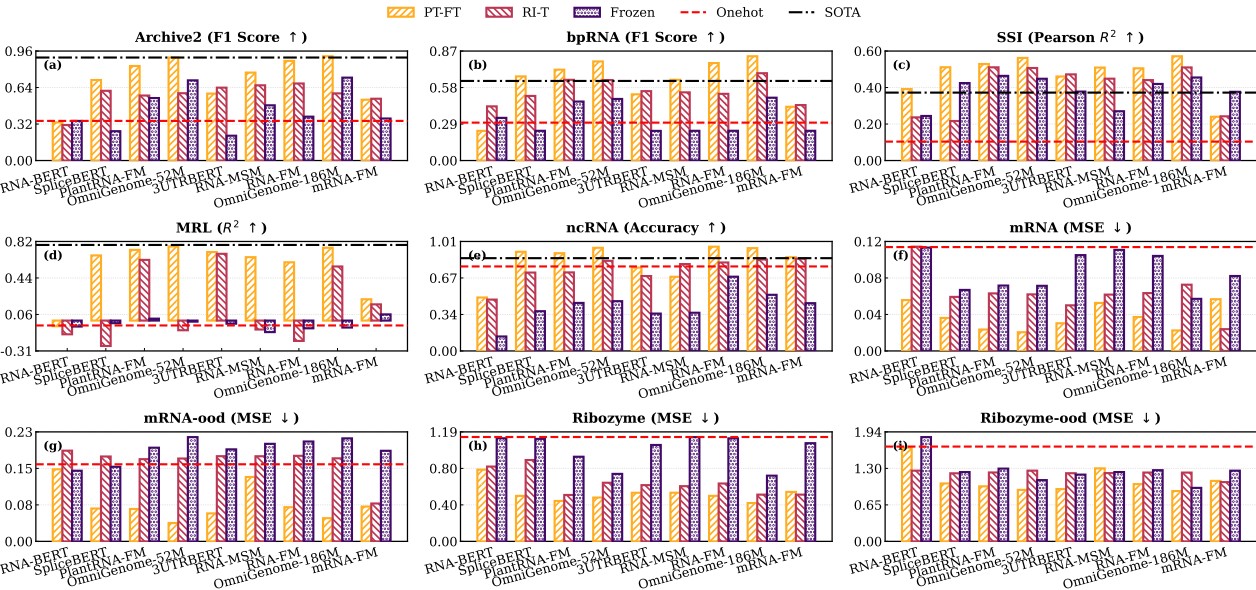

*Figure 1.* For each task, the orange bars (`PT-FT`) represent the performance of RNA foundation models fine-tuned from pretrained weights, while the red bars (`RI-T`) correspond to models trained from random initialization on downstream tasks. The purple bars (`Frozen`) denote performance obtained by applying linear probing to frozen last-layer representations of pretrained RNA foundation models. The red dashed lines indicate results achieved using one-hot sequence representations (`Onehot`), and the black dash–dot lines represent previously reported state-of-the-art (`SOTA`) performance. Across most tasks, fine-tuning from pretrained weights consistently outperforms the `Onehot` baseline. In particular, models such as `OmniGenome-52M` and `OmniGenome-186M` achieve performance that is consistently superior to, or at least comparable with, the reported `SOTA` results across all nine evaluated tasks.

loss landscape onto a two-dimensional subspace. This subspace is spanned by two randomly sampled, normalized direction vectors, $\delta$ and $\eta$, and is centered at the final parameter configuration $\theta^*$. The loss $\mathcal{L}$ is evaluated over this plane as a function of two scalar coefficients, $\alpha$ and $\beta$:

$$f(\alpha, \beta) = \mathcal{L}(\theta^* + \alpha\delta + \beta\eta). \quad (1)$$

Importantly, we adopt the filter-wise normalization scheme introduced in the original work. This normalization scales the perturbation applied to each filter in proportion to the norm of its corresponding weights, thereby preventing the visualization from being dominated by parameters with large magnitudes. As a result, the visualization becomes approximately scale-invariant, enabling fair comparisons of the geometric properties of the loss surface, particularly the *flatness* and *width* of the loss basins. By comparing the loss landscapes of models trained from different initialization, we assess whether pretraining systematically biases optimization toward wider minima, which are commonly associated with improved generalization. We additionally compute quantitative flatness indicators using a SAM-style perturbation metric (Foret et al., 2020). Lower values indicate flatter local neighborhoods around the converged solution. These quantitative flatness results are reported in Appendix Table 6.

## 5. Results

We begin by characterizing the overall transfer learning behavior across nine downstream tasks using four training settings: `Onehot`, `Frozen`, `RI-T`, and `PT-FT`. Figure 1 summarizes the results, and detailed results can be found in Appendix Table 7, Table 8, and Table 9. Across almost all tasks and models, `PT-FT` achieves the best or near-best performance. In contrast, `Frozen` representations show highly task-dependent behavior and frequently underperform `PT-FT`. Notably, `Onehot` remains competitive on certain tasks, such as ncRNA classification, indicating that strong downstream performance does not universally require pretrained representations.

These observations raise a central question: What mechanisms underlie the performance gains of fine-tuning, given that frozen representations are often insufficient? We structured our experiments and analysis around the following three hypotheses.

- ***Hypothesis 1.*** The pretraining objective aligns well with downstream objective, thus using frozen last layer representation should yield good performance.

- ***Hypothesis 2.*** Finetuning achieves better performance than training from scratch as there exists low-level feature reuse, or hierarchical feature reuse.

- **Hypothesis 3.** Finetuning achieves better performance as they start training from better initial weights.

### 5.1. Experimental Results for Hypothesis 1

**Limitations of Linear Probing on Frozen Representations of RNA Foundation Models** Our investigation into frozen features reveals a striking dichotomy in their effectiveness, directly challenging the notion of a universally powerful feature extractor. The performance landscape, as depicted in Figure 1, is highly task-dependent. For instance, in structure prediction tasks such as `Archive2` and `bpRNA`, `Frozen` representations from models like `OmniGenome-52M` and `PlantRNA-FM` consistently and substantially outperform the `Onehot` baseline. This demonstrates a clear scenario where pre-trained features provide a significant advantage.

Conversely, on the `ncRNA` classification task, the trend reverses. The `Onehot` baseline achieves a remarkable accuracy of nearly 80%, surpassing most frozen models, some by a large margin. This counterintuitive result suggests that the `ncRNA` task may be heavily reliant on distinct, local sequence statistics that define each RNA class. A simple `Onehot` encoding perfectly captures this information, while the abstract, high-level features learned via MLM pre-training might obscure these critical local signals or introduce irrelevant contextual noise.

This heterogeneity leads to a crucial insight: the value of frozen features is a direct function of alignment. The success on SSP tasks is likely because the pre-training objective implicitly or explicitly learns structural properties that are directly transferable. The failure on `ncRNA` highlights a misalignment, where the pre-trained inductive biases do not match the problem's underlying statistical nature. Therefore, evaluating a model's off-the-shelf capability requires a nuanced, task-centric approach rather than a blanket judgment.

**The Inadequacy of Final-Layer Features Compared to Full Fine-Tuning.** Even for tasks in which frozen representations outperform baseline methods (e.g., `Archive2` and `SSI`), a substantial performance gap remains relative to full fine-tuning. This result indicates that, although final-layer representations encode useful information, they are insufficient for achieving optimal performance. Effective adaptation to new data distributions and task objectives requires adjustments across the model's feature hierarchy rather than reliance on fixed high-level features alone. By enabling parameter updates throughout the network, fine-tuning allows the model to move beyond its pretrained solution manifold toward more task-specialized representations, underscoring the importance of deep, task-specific feature adaptation. To test whether this conclusion is specific to final-layer probing, we additionally evaluated frozen probes at representative intermediate depths ($L/4$, $L/2$, and $3L/4$) in Appendix Table 5. Intermediate layers can outperform the final layer in several cases, indicating that final-layer-only probing can underestimate frozen-feature utility. Nevertheless, coarse depth selection does not overturn the overall advantage of full fine-tuning on the tested tasks.

> **Finding 1:** The effectiveness of frozen representations from RNA foundation models is task-dependent and insufficient for optimal transfer. When pretraining objectives align with downstream tasks, frozen features can outperform baseline methods; however, they consistently lag behind full fine-tuning. This indicates that while frozen representations can contain useful information at different depths, effective adaptation generally requires coordinated updates across the model's feature hierarchy.

### 5.2. Experimental Results for Hypothesis 2

Given that the results for Hypothesis 1 indicate that full fine-tuning is often necessary, a natural follow-up question concerns how fine-tuning facilitates effective transfer. By analyzing the block-wise $\ell_2$ distance from pretrained weights throughout fine-tuning (Figures 2, 3, and 6), we observe that the classical computer vision paradigm of hierarchical feature reuse does not consistently apply across RNA tasks and model architectures. Instead, our analysis reveals three distinct patterns of knowledge transfer. This reinforces the view that fine-tuning induces heterogeneous adaptation trajectories rather than a universal hierarchy of feature reuse.

**Aligned Feature Reuse.** The secondary structure prediction (SSP) tasks provide a canonical example of effective feature reuse. As shown in Figure 2, models such as `OmniGenome-52M` and `OmniGenome-186M`—both pretrained with explicit secondary structure prediction objectives—exhibit the characteristic behavior of hierarchical reuse. In particular, their shallow layers remain highly stable throughout fine-tuning, as evidenced by the deep blue regions in the block-wise $\ell_2$ distance heatmaps. This stability, together with their strong downstream performance (Figures 1(a) and Figure 1(b)), indicates that the low-level representations learned during pretraining are well aligned with the SSP task and can be directly reused with minimal task-specific adaptation.

**Feature Stagnation:** A more revealing case is observed for `RNA-BERT` on the same SSP tasks. As shown in Figure 2, its shallow layers remain largely stable during fine-tuning, a pattern that could be mistakenly interpreted as evidence of effective feature reuse. However, Figure 1(a) demonstrates that `RNA-BERT` exhibits markedly poor performance, even under performing models trained from random initializa-

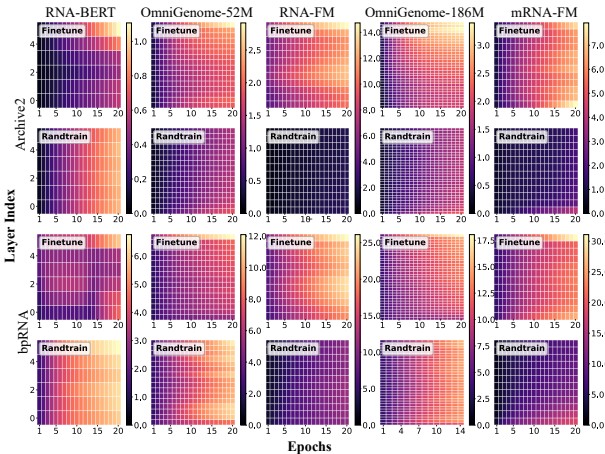

*Figure 2.* Block-wise $l_2$ distance evolution during training. Heatmaps depict the evolution of the $\ell_2$ distance between model parameters and their initial values for each transformer block over 20 training epochs. Results are shown for five models evaluated on the **Archive2** (top two rows) and **bpRNA** (bottom two rows) tasks under both `PT-FT` and `RI-T` settings. For each model–task pair, the color scale is shared between two training settings, with warmer colors indicating larger deviations from the initial weights.

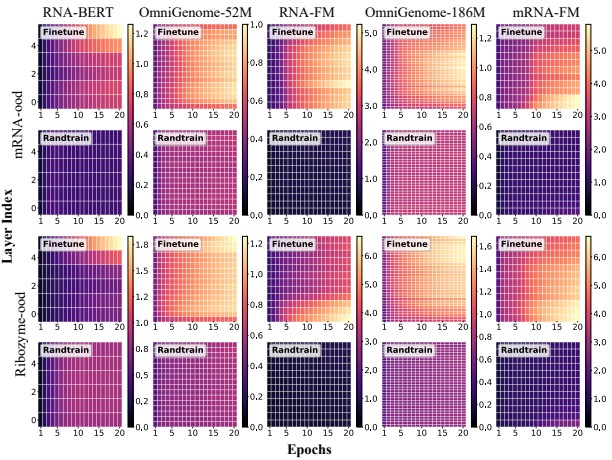

*Figure 3.* Block-wise $l_2$ distance evolution during training on the **mRNA-ood** (top two rows) and **Ribozyme-ood** (bottom two rows) tasks, under both `PT-FT` and `RI-T` settings.

tion. This behavior reflects feature stagnation rather than productive reuse: the pretrained representations are fundamentally misaligned with the SSP task and fail to adapt during fine-tuning. As a result, the apparent stability of these layers constitutes a detrimental constraint, leading to negative transfer instead of performance gains.

**Global Adaptation:** For function prediction tasks where pretraining–task alignment is not explicit (Figures 3 and Figures 6), we observe a markedly different transfer pattern. In the two out-of-distribution settings, `PT-FT` induces strong, global parameter updates across all layers, as evidenced by widespread yellow and red regions in the block-wise distance heatmaps. This behavior indicates that the pretrained feature hierarchy is largely overwritten rather than reused. Instead of preserving specific representations, the models primarily exploit pretrained weights as a superior initialization, enabling comprehensive re-specialization of the representational space to accommodate the new task.

**Contrast with Computer Vision Paradigms.** Collectively, these findings indicate that the principle of hierarchical feature reuse does not generally extend to RNA foundation models. In contrast to CNNs in computer vision, where early layers often learn broadly transferable features, the token-level embeddings and local attention patterns learned by RNA models appear to be substantially less invariant under adaptation to specific downstream tasks. The pronounced parameter updates observed in early layers suggest that, rather than simply leveraging a fixed, pretrained syntactic representation of sequences, RNA models must

re-contextualize their fundamental input representations to accommodate task-specific requirements.

> *Finding 2*: The hierarchical feature reuse paradigm established in computer vision does not generalize to RNA FMs. Instead, effective transfer is governed by the alignment between pretraining objectives and downstream task requirements, determining whether pretrained representations are reused, stagnate, or undergo global reconfiguration during fine-tuning.

### 5.3. Experimental Results for Hypothesis 3

Although `PT-FT` consistently outperforms `RI-T`, this advantage cannot be explained by the preservation or reuse of shallow features across tasks. Instead, Hypothesis 3 posits that the primary benefit of pretraining lies in providing a favorable initialization that facilitates optimization. Such an initialization is expected to accelerate convergence and guide the model toward solutions located in wider and flatter regions of the loss landscape, which are usually associated with improved generalization. We test this hypothesis by analyzing training dynamics and visualizing the geometry of the loss surface.

**Faster Convergence through Better Initialization.** The training loss curves in Figure 4 provide evidence of accelerated convergence under fine-tuning. Across nearly all tasks and model configurations, `PT-FT` models converge more rapidly to lower loss values than that trained from random initialization. These observations support the hypothesis that pretraining places the model in a favorable region of the parameter space, from which optimization proceeds more directly toward high-quality solutions.

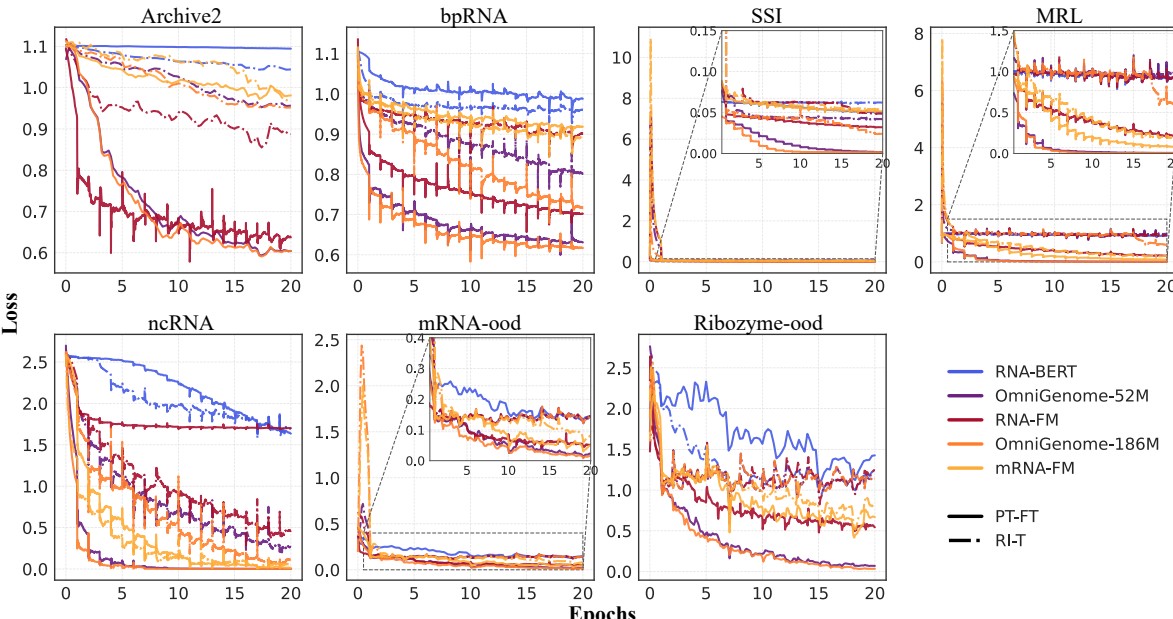

*Figure 4.* Training loss curves of RNA foundation models under `PT-FT` and `RI-T` settings across seven tasks. Solid lines represent fine-tuning from pretrained weights (`PT-FT`), and dashed lines represent training from randomly initialized weights (`RI-T`).

**Failure Case: Insufficient Model Capacity (Figure 7).** A notable exception is `RNA-BERT`. The loss landscapes of `RNA-BERT` are consistent with its inferior empirical performance. Both `PT-FT` and `RI-T` variants converge to sharp and highly irregular minima, indicating unstable optimization behavior. This phenomenon can be primarily attributed to the model's limited capacity, as `RNA-BERT` contains only 0.5M parameters. Such a small model lacks sufficient representational power to extract robust and transferable features during pre-training, and consequently fails to induce a smooth and well-structured optimization trajectory for downstream tasks. As a result, fine-tuning offers little advantage over training from scratch, leading to similarly chaotic and ineffective convergence behavior.

Importantly, these observations do not suggest that larger model capacity necessarily leads to improved transfer performance. For example, `mRNA-FM` (239M parameters) exhibits comparable performance under both the `PT-FT` and `RI-T` settings across all evaluated tasks. As shown in Figures 3 and Figure 6, its model parameters undergo substantial deviations from their initial values during fine-tuning, particularly in the lower layers. This indicates limited retention of pretrained representations. Moreover, the loss curves in Figure 4 show that `mRNA-FM` fails to converge to a stable solution. This behavior is likely related to the increased optimization difficulty of large Transformer models when trained or fine-tuned on small datasets, where over-parameterization may amplify training instability rather than facilitate convergence.

**Successful Convergence to Flat Minima (Figure 5).** On the SSP tasks, models such as `OmniGenome-52M`, `OmniGenome-186M`, and `RNA-FM` show clear benefits from pretraining. After fine-tuning, these models converge to solutions associated with wider and flatter minima. In contrast, models trained from random initialization typically converge to sharper minima, even when achieving comparable loss values.

**Quantitative Flatness Analysis.** We additionally evaluated the SAM-style perturbation around the converged solutions (Appendix Table 6). On Archive2, ncRNA and mRNA-ood tasks, `PT-FT` achieves lower SAM-style metrics in all 15 tested model-task pairs. These quantitative results support a task-conditional interpretation of pretraining as an optimization prior.

Overall, the results indicate that in many tested settings, especially structure-aligned ones, pretraining benefits downstream learning by providing a favorable optimization initialization. Importantly, this benefit is contingent on the alignment between the pretraining objective and the downstream task. When such alignment is strong, pretraining positions the model in a region of the parameter space that facilitates faster convergence and leads to flatter minima in the loss landscape, as observed for the SSP tasks. In contrast, weak task relatedness results in substantial overwriting of pretrained representations, yielding optimization behavior similar to training from random initialization. This effect is further influenced by model capacity: insufficient capacity limits the acquisition of transferable inductive biases,

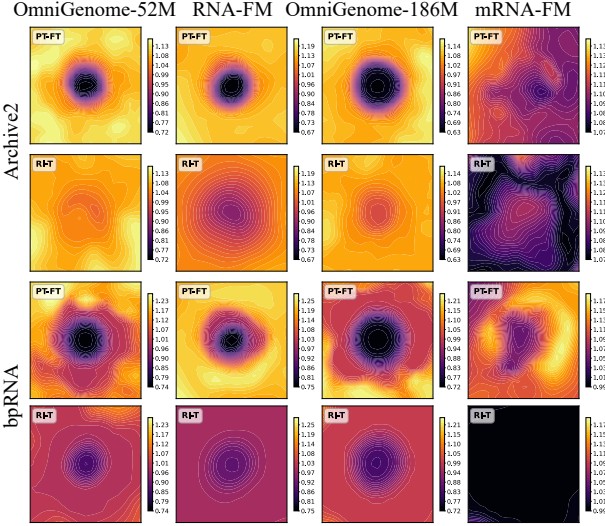

*Figure 5.* Pre-training facilitates flatter minima in secondary structure prediction tasks. Loss landscape visualizations for **Archive2** and **bpRNA** across `OmniGenome-52M`, `RNA-FM`, and `OmniGenome-186M` show that fine-tuning from pretrained weights (`PT-FT`) consistently converges to wider and smoother basins. In contrast, models trained from random initialization (`RI-T`) tend to converge to sharper or less well-defined minima. These results indicate that pretraining introduces structure-relevant inductive biases that reshape the optimization landscape, leading to more stable solutions with improved generalization compared to training from scratch.

whereas overly large models fine-tuned on small datasets may experience optimization instability.

> ***Finding* 3**: In many tested settings, pretraining serves as an optimization prior, whose effectiveness depends on both task alignment and model capacity. When the pretraining objective is well aligned with the downstream task and model capacity is sufficient, pretraining accelerates convergence and guides optimization toward flatter minima. In contrast, when such alignment is weak or capacity is mismatched, the advantage of pretraining over random initialization diminishes due to extensive representation overwriting or optimization instability.

## 6. Discussion

**Conclusion.** This work challenges the assumption that current RNA foundation models function as universal feature extractors. Instead, our results demonstrate that the effectiveness of pretrained representations is strongly governed by the alignment between the pretraining objective and the downstream task. When such alignment exists, frozen representations can provide strong baselines and require only limited adaptation in top layers.

More broadly, the hierarchical feature reuse paradigm often observed in computer vision does not consistently hold for the RNA FMs evaluated in this study. In lower-capacity models, pretrained initialization may restrict effective adaptation, whereas in larger models applied to weakly aligned tasks, extensive parameter changes during fine-tuning suggest that pretrained representations are substantially overwritten rather than directly reused. By contrast, in structure-aligned settings, pretraining can provide a favorable initialization associated with flatter solutions and improved generalization. These findings indicate that effective transfer depends on the interaction between task alignment and model capacity, rather than on increased model scale alone.

**Implications and Limitations.** Taken together, these findings suggest that the RNA FMs evaluated in this study do not yet provide uniformly reusable, task-agnostic representations across diverse downstream settings. Instead, their transfer performance depends on the interplay between task alignment, model capacity, and optimization dynamics, suggesting that more general-purpose RNA FMs may require improved pretraining objectives and evaluation protocols. In particular, for fitness prediction tasks, future evaluations should account for properties of the underlying fitness landscapes, since variation in landscape topology may affect task difficulty and complicate the interpretation of relative model performance (Huang et al., 2025). Simply scaling model size under standard MLM objectives may offer limited benefits for many downstream RNA tasks.

From a practical perspective, our results indicate that parameter adaptation is often required for tasks that are weakly related to the pretraining objective. However, whether full fine-tuning is necessary, or whether more parameter-efficient adaptation strategies can achieve better performance, remains an important question. In terms of scope, our analysis primarily focuses on small- to medium-sized, encoder-only Transformer architectures. The conclusions may not directly extend to substantially larger models or models based on alternative architectural or pretraining paradigms.

Our intermediate-layer probing analysis shows that frozen representations from non-final layers can outperform final-layer probing in some cases, suggesting that transferable knowledge may be distributed across model depth. However, improved probing performance indicates downstream utility rather than biological interpretability. Future work should therefore move beyond performance-based layer comparisons and develop systematically validated interpretability protocols to determine whether, where, and under what task settings RNA FMs encode biologically meaningful determinants of RNA structure and function (Zhou et al., 2026). Together, these directions are necessary for distinguishing predictive utility from genuinely reusable and biologically informative representation learning in RNA FMs.

## Acknowledgements

This work was supported by the UKRI Future Leaders Fellowship under Grant MR/S017062/1 and MR/X011135/1; in part by National Natural Science Foundation of China under Grant 62376056 and 62076056; in part by the Isambard-AI, Royal Society Faraday Discovery Fellowship (FDF/S2/251014), BBSRC Transformative Research Technologies (UKRI1875), Royal Society International Exchanges Award (IES/R3/243136), Kan Tong Po Fellowship (KTP/R1/231017); and the Amazon Research Award and Alan Turing Fellowship.

## Impact Statement

This study examines the mechanisms underlying transfer learning in RNA foundation models and provides empirical insights into the conditions under which pretraining is effective. Our results indicate that current RNA foundation models do not yet learn universally transferable representations, emphasizing the importance of task alignment and optimization behavior. These findings contribute to a more precise understanding of model transferability and may inform future research on pretraining strategies and evaluation practices in RNA sequence modeling.

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

# A. Methods

## A.1. Experimental Settings

**Experimental Platform.** All experiments were conducted using the OmniGenBench (v0.4.8a0), which provides unified implementations for model loading, dataset processing, training, and evaluation across diverse genomic tasks. Training was performed using the `AccelerateTrainer` to standardize the training implementation across models and tasks. All experiments were executed on $4\times$ NVIDIA H100 GPUs.

**Pretrained Models and Datasets.** We conducted experiments based on a diverse suite of downstream tasks, and RNA foundation models of different sizes, as shown in Tables 2 and 3.

*Table 2.* **Overview of RNA tasks and datasets.**

| RNA Task | Metric | Task Type | #Train/Val/Test | Min/Max Length | Source/Venue |
|---|---|---|---|---|---|
| **Structure Prediction** | | | | | |
| SSP - Archive2 (small data regime) | $ACC$ | Token Cls | 608/76/76 | 151/500 | OmniGenBench |
| SSP - bpRNA | $ACC$ | Token Cls | 10,814/1,300/1,305 | 134/512 | BEACON |
| SSI | Pearson $R^2$ | Token Reg | 14,049/1,756/3,095 | 100/100 | BEACON |
| **Function Prediction** | | | | | |
| ncRNA | $ACC$ | Sequence Cls | 5,670/650/2,600 | 38/1182 | BEACON |
| MRL | $R^2$ | Sequence Reg | 9,000/3,000/3,000 | 25/100 | BEACON |
| Ribozyme | MSE | Sequence Reg | 38,058/12,686/12,686 | 425/425 | RNAGym |
| Ribozyme-ood (fewer training samples) | MSE | Sequence Reg | 1413/354/61,663 | 425/425 | RNAGym |
| mRNA | MSE | Sequence Reg | 4753/1584/1585 | 282/282 | RNAGym |
| mRNA-ood (fewer training samples) | MSE | Sequence Reg | 1147/287/6488 | 282/282 | RNAGym |

*Table 3.* **Overview of RNA foundation models.**

| Model | #Params | Tokenization | Pre-train Size | Data Source | Species |
|---|---|---|---|---|---|
| RNA-FM | 99M | SNT | 23M seqs | RNA central | Multi-species |
| RNA-MSM | 96M | SNT | 4096 families | Rfam (MSA) | Multi-species |
| RNA-BERT | 0.5M | SNT | 4096 families | Rfam | Multi-species |
| SpliceBERT | 19M | SNT | 2M seqs | UCSC pre-mRNA | Multi-species |
| 3UTRBERT | 86M | 3-mer | 20k seqs | GENCODE UTR | Human |
| mRNA-FM | 239M | 3-mer | 45M seqs | mRNA Coding Seqs | Multi-species |
| PlantRNA-FM | 34M | SNT | 54.2B | OneKP | 1124 plants |
| OmniGenome-52M | 52M | SNT | 54.2B tokens | OneKP | 1124 plants |
| OmniGenome-186M | 186M | SNT | 54.2B tokens | OneKP | 1124 plants |

**Training Settings.** To ensure a fair comparison across transfer settings, we adopted consistent optimization hyperparameters within each training regime. For fine-tuning from pretrained weights (`PT-FT`) and frozen-representation training (`Frozen`), the learning rate was set to $2 \times 10^{-5}$. For training from random initialization (`RI-T`), a higher learning rate of $1 \times 10^{-4}$ was used to facilitate optimization from scratch. All models were trained for 20 epochs using the AdamW optimizer. Batch sizes were task-dependent and fixed across models within each task (e.g., 32 for Archive2, ncRNA and SSI, 64 for MRL, and 16 for long-sequence bpRNA tasks). All experiments were repeated using three different random seeds (2025, 2026, and 2027), and consistent trends were observed across runs.

The unified training protocol was chosen to maximize cross-model comparability rather than to claim per-model optimal hyperparameter tuning. To examine whether the main `PT-FT` versus `RI-T` comparison is an artifact of a single default learning-rate choice, we conducted a targeted learning-rate sensitivity analysis on ncRNA classification and mRNA-ood fitness prediction tasks. We evaluated OmniGenome-52M, OmniGenome-186M, RNA-BERT, RNA-FM, and mRNA-FM with `PT-FT` learning rates $\{1e-5, 2e-5, 5e-5\}$ and `RI-T` learning rates $\{1e-3, 1e-4, 1e-5\}$, averaging results over three random seeds. As summarized in Table 4, the best `PT-FT` result remains stronger than the best `RI-T` result for all

tested model-task pairs. The results also show that `RI-T` is more sensitive to learning-rate choice than `PT-FT`, indicating that per-model tuning can affect absolute gaps but does not overturn the relative advantage of pretrained initialization in the tested settings.

*Table 4.* **Learning-rate sensitivity analysis.** Results are averaged over three random seeds. Within each task-model row, the best learning rate for each training setting is marked in bold type.

| Task | Model | PT-FT | | | RI-T | | |
|---|---|---|---|---|---|---|---|
| | | LR=1e-5 | LR=2e-5 | LR=5e-5 | LR=1e-3 | LR=1e-4 | LR=1e-5 |
| ncRNA ↑ | OmniGenome-52M | 0.9334 | **0.9403** | **0.9403** | 0.7652 | 0.8215 | **0.8332** |
| | OmniGenome-186M | 0.9407 | **0.9496** | 0.9334 | 0.7960 | 0.8325 | **0.8426** |
| | RNA-BERT | 0.3176 | 0.4719 | **0.5973** | 0.3024 | **0.4452** | 0.4031 |
| | RNA-FM | 0.9350 | 0.9496 | **0.9632** | 0.6531 | **0.8099** | 0.7901 |
| | mRNA-FM | 0.8323 | **0.8559** | 0.8543 | 0.7932 | 0.8337 | **0.8438** |
| mRNA-ood ↓ | OmniGenome-52M | 0.0465 | 0.0370 | **0.0356** | **0.1950** | 0.1989 | 0.2019 |
| | OmniGenome-186M | 0.0371 | **0.0361** | 0.0394 | 0.2022 | 0.1999 | **0.1939** |
| | RNA-BERT | 0.1643 | 0.1342 | **0.1278** | **0.1434** | 0.2026 | 0.1857 |
| | RNA-FM | 0.1227 | 0.0735 | **0.0571** | **0.1057** | 0.2017 | 0.2001 |
| | mRNA-FM | 0.0925 | 0.0721 | **0.0662** | 0.0882 | **0.0769** | 0.0804 |

**Task Heads and Objective Functions.** Unless otherwise specified, we employed standardized OmniGenome task heads that operate on the final hidden states of the backbone encoder.

For training settings including `RI-T`, `PT-FT`, and `Frozen`, we used identical task heads, loss functions, and optimization procedures. The three settings differ only in how the backbone parameters are initialized and updated: in `RI-T`, all model parameters are randomly initialized and trained from scratch; in `PT-FT`, all parameters are initialized from pretrained weights and jointly fine-tuned; and in `Frozen`, pretrained backbone parameters are kept fixed while only the task head is trained. This design ensures that performance differences across settings are attributable solely to transfer learning effects rather than architectural or objective mismatches.

For secondary structure prediction (SSP), we used `OmniModelForTokenClassification`, which applies a linear classifier followed by a softmax function to each token representation from the final hidden layer, $\text{Linear}(\mathbf{h}_t) \to \text{Softmax}$. Models are trained using cross-entropy loss over token labels, with padding positions ignored via mask (encoded as $-100$).

For ncRNA family classification, we used `OmniModelForSequenceClassification`, which aggregates token representations into a sequence-level vector using `OmniPooling`, followed by a linear classifier and softmax. Training is performed using cross-entropy loss on sequence-level labels.

For mean ribosome loading (MRL) and all fitness prediction tasks, we used `OmniModelForSequenceRegression`. In this setting, `OmniPooling` is applied to obtain a sequence-level representation, which is then passed through a linear regression head. The training objective is mean squared error (MSE), and label positions marked with an ignore value (default $-100$) are excluded from loss computation.

For structural score imputation (SSI), final hidden states are first mapped to nucleotide-level resolution in a tokenizer-dependent manner: (i) direct alignment for single-nucleotide tokenization, (ii) $k$-mer expansion for $k$-mer tokenization. Mapped representations are filtered using a provided `weight_mask`, projected through a lightweight MLP, concatenated with an embedding of the observed structural scores, and passed to a linear regressor to predict missing structural values. The SSI objective is an MSE loss computed only over valid target positions, with ignored positions masked out (e.g., $-100$).

**One-hot baselines.** We implemented task-matched one-hot baselines to quantify the performance of simple sequence statistics without pretrained representations. For all one-hot baselines, nucleotide sequences were first mapped to integer indices and then converted to one-hot vectors at runtime. All models were lightweight linear predictors trained with Adam/AdamW and selected using the validation set; the test score reported corresponds to the checkpoint achieving the best validation performance. Unless otherwise stated, experiments were repeated with three random seeds and trained for 50 epochs. Padding was applied to handle variable-length sequences, and padded labels were masked out when computing losses/metrics (e.g., $-100$ for token-level tasks).

## A.2. Additional Results and Analyses

**Intermediate-layer Frozen Probing.** To examine whether final-layer probing underestimates the utility of frozen representations, we performed additional frozen probing at four representative depths: $L/4$, $L/2$, $3L/4$, and the final layer. As shown in Table 5, intermediate layers can outperform the final layer, confirming that final-layer-only probing is not always the strongest frozen-feature baseline. However, selecting among these coarse layer depths does not overturn the empirical advantage of full fine-tuning in the tested settings.

*Table 5.* **Intermediate-layer frozen probing analysis.** Coarse frozen probing at $L/4$, $L/2$, $3L/4$, and the final layer shows that intermediate layers can improve over the final layer, but the best frozen depth does not overturn the `PT-FT` advantage in the tested settings. The best result in each row is marked in bold type.

| Task | Model | $L/4$ | $L/2$ | $3L/4$ | **Final Layer** |
|---|---|---|---|---|---|
| Archive2 ↑ | OmniGenome-52M | $0.440 \pm 0.018$ | $0.577 \pm 0.010$ | $0.646 \pm 0.008$ | $\mathbf{0.721 \pm 0.001}$ |
| | OmniGenome-186M | $0.547 \pm 0.005$ | $0.614 \pm 0.006$ | $0.669 \pm 0.005$ | $\mathbf{0.726 \pm 0.001}$ |
| | RNA-BERT | $0.296 \pm 0.038$ | $0.299 \pm 0.022$ | $0.293 \pm 0.026$ | $\mathbf{0.335 \pm 0.039}$ |
| | RNA-FM | $0.602 \pm 0.010$ | $\mathbf{0.626 \pm 0.011}$ | $0.617 \pm 0.012$ | $0.377 \pm 0.031$ |
| | mRNA-FM | $0.473 \pm 0.005$ | $\mathbf{0.476 \pm 0.004}$ | $\mathbf{0.476 \pm 0.004}$ | $0.350 \pm 0.001$ |
| ncRNA ↑ | OmniGenome-52M | $0.428 \pm 0.057$ | $\mathbf{0.605 \pm 0.006}$ | $0.568 \pm 0.019$ | $0.486 \pm 0.002$ |
| | OmniGenome-186M | $0.618 \pm 0.004$ | $\mathbf{0.751 \pm 0.004}$ | $0.711 \pm 0.009$ | $0.525 \pm 0.007$ |
| | RNA-BERT | $\mathbf{0.260 \pm 0.026}$ | $0.116 \pm 0.054$ | $0.096 \pm 0.028$ | $0.132 \pm 0.005$ |
| | RNA-FM | $0.676 \pm 0.037$ | $\mathbf{0.763 \pm 0.004}$ | $0.755 \pm 0.010$ | $0.707 \pm 0.004$ |
| | mRNA-FM | $\mathbf{0.588 \pm 0.004}$ | $0.521 \pm 0.039$ | $0.532 \pm 0.034$ | $0.424 \pm 0.047$ |
| mRNA-ood ↓ | OmniGenome-52M | $0.197 \pm 0.004$ | $0.214 \pm 0.010$ | $\mathbf{0.188 \pm 0.004}$ | $0.190 \pm 0.005$ |
| | OmniGenome-186M | $0.173 \pm 0.013$ | $0.167 \pm 0.009$ | $\mathbf{0.106 \pm 0.005}$ | $0.199 \pm 0.020$ |
| | RNA-BERT | $0.180 \pm 0.001$ | $0.168 \pm 0.008$ | $0.179 \pm 0.002$ | $\mathbf{0.151 \pm 0.000}$ |
| | RNA-FM | $0.194 \pm 0.002$ | $0.193 \pm 0.007$ | $\mathbf{0.191 \pm 0.007}$ | $0.195 \pm 0.003$ |
| | mRNA-FM | $\mathbf{0.129 \pm 0.015}$ | $0.154 \pm 0.010$ | $0.154 \pm 0.025$ | $0.174 \pm 0.004$ |

**Local Loss Sensitivity Analysis.** Because two-dimensional loss-landscape visualizations provide only qualitative evidence, we additionally report base loss and a SAM-style perturbation-induced loss increase ($\Delta L_{\text{SAM}}$). Lower values indicate lower measured local loss sensitivity under the SAM diagnostic. To avoid confounding from architecture- and parameterisation-dependent scale effects, we interpret these quantities through paired comparisons between `PT-FT` and `RI-T` within the same backbone and task, rather than as absolute comparisons across different model architectures. Table 6 shows that `PT-FT` yields lower SAM-metric values than `RI-T` for every reported model-task pair.

*Table 6.* **Local loss sensitivity analysis (base loss and SAM metric).** The better value in each row is marked in bold type.

| Task | Model | PT-FT | | RI-T | |
|---|---|---|---|---|---|
| | | **Base Loss** | **SAM Metric** | **Base Loss** | **SAM Metric** |
| Archive2 ↓ | OmniGenome-52M | **0.5764** | **0.0011** | 0.5832 | 0.0039 |
| | OmniGenome-186M | **0.5871** | **0.0023** | 0.6801 | 0.0043 |
| | RNA-FM | **0.5799** | **0.0012** | 0.9240 | 0.0034 |
| | mRNA-FM | **0.7934** | **0.0214** | 0.7951 | 0.0288 |
| ncRNA ↓ | OmniGenome-52M | **1.6962** | **0.0088** | 2.1012 | 0.0142 |
| | OmniGenome-186M | **1.6925** | **0.0006** | 1.9670 | 0.0206 |
| | RNA-BERT | 2.2527 | **0.0223** | **2.0121** | 0.0233 |
| | RNA-FM | **1.8789** | **0.0540** | 2.2900 | 0.0560 |
| | mRNA-FM | **1.8746** | **0.0533** | 2.2775 | 0.0838 |
| mRNA-ood ↓ | OmniGenome-52M | **0.0138** | **0.2531** | 0.1364 | 0.3197 |
| | OmniGenome-186M | **0.0089** | **0.2142** | 0.1491 | 0.4744 |
| | RNA-BERT | **0.2225** | **0.1208** | 0.2356 | 0.2958 |
| | RNA-FM | **0.1007** | **0.1459** | 0.1338 | 0.4652 |
| | mRNA-FM | 0.1376 | **0.0713** | **0.1329** | 0.7774 |

# B. Additional Tables and Figures

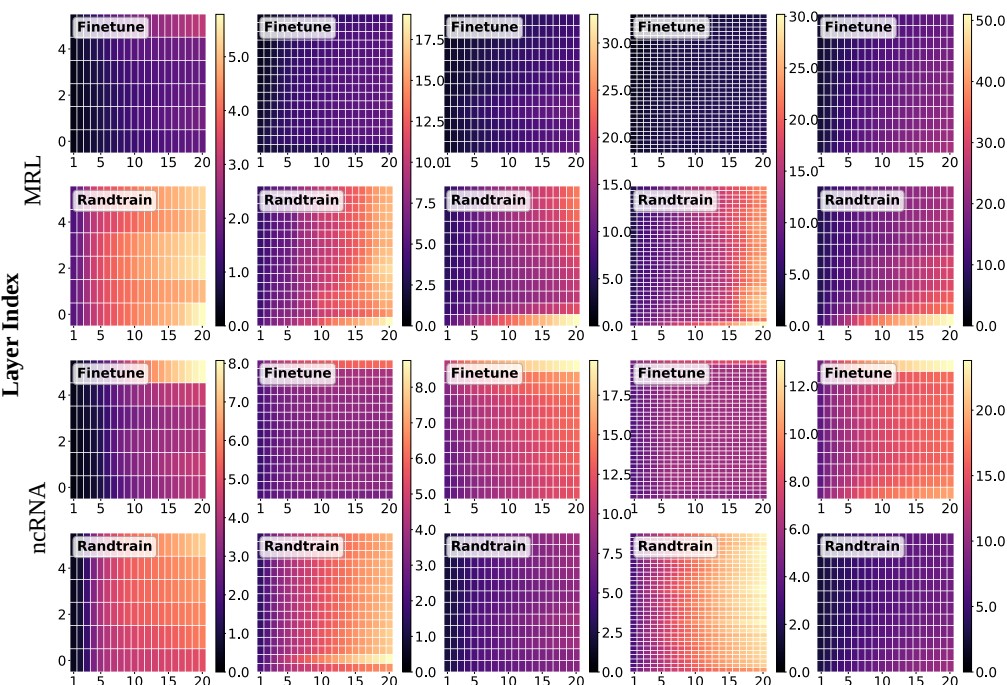

*Figure 6.* Block-wise $l_2$ distance evolution during training on the **MRL** (top two rows) and **ncRNA** (bottom two rows) tasks, under both `PT-FT` and `RI-T` settings. The horizontal axis corresponds to training epochs, while the vertical axis indexes the transformer hidden layers. Each heatmap shows the $\ell_2$ distance between the parameters of each hidden layer and their initial values throughout training.

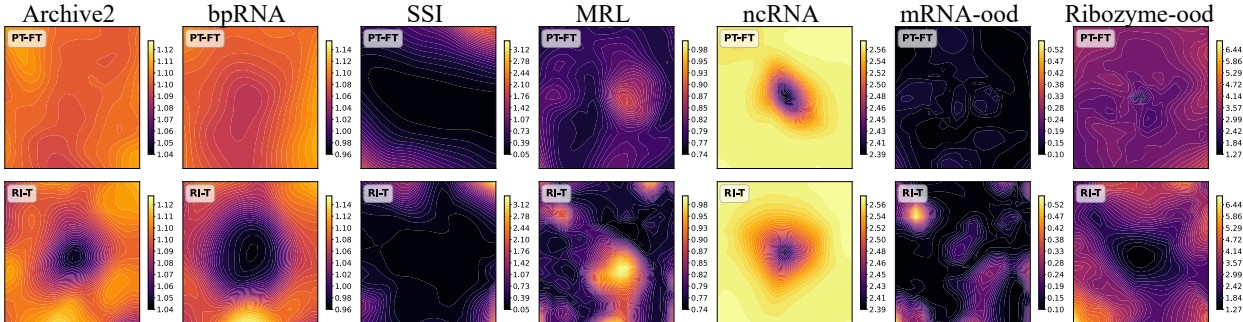

*Figure 7.* Failure Case of Insufficient Capacity. Visualization of the loss landscapes for `RNA-BERT` across all downstream tasks. On most cases, both `PT-FT` (top row) and `RI-T` (bottom row) models converge to sharp, irregular minima characterized by rugged topologies. On `bpRNA` task, models can find a better local minima under `RI-T` training settings.

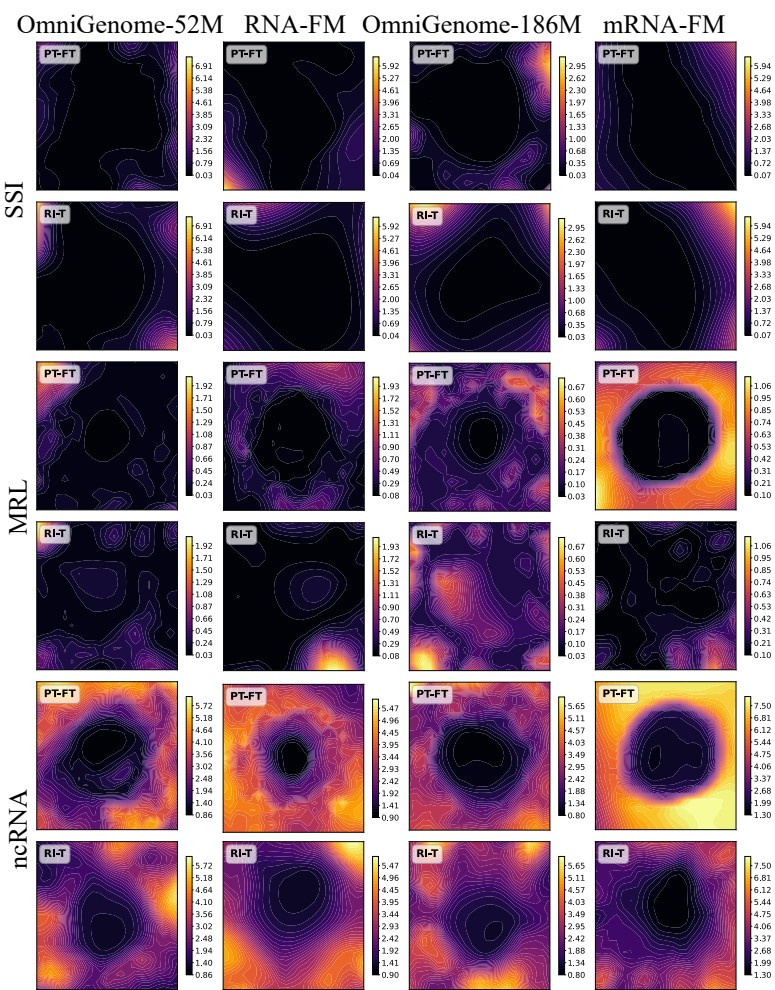

*Figure 8.* Loss landscape visualizations under `PT-FT` and `RI-T`. Columns correspond to four models (`OmniGenome-52M`, `RNA-FM`, `OmniGenome-186M`, and `mRNA-FM`). Rows correspond to three downstream tasks (**SSI**, **MRL**, and **ncRNA**). For each task, the top sub-row shows fine-tuning from pretrained weights (`PT-FT`) and the bottom sub-row shows training from random initialization (`RI-T`). Each panel plots the loss surface in a 2D subspace centered at the converged solution using the projection method of Li et al. (Li et al., 2018a). Darker regions indicate lower loss and contour lines depict local geometry. Color scales are shown per panel for reference.

*Table 7.* **Summary of results using different training settings on downstream tasks(Part 1). Black bold values** indicate underperformance relative to the `Onehot` baseline. **Orange bold values** indicate that `PT-FT` outperforms `RI-T`.

| Task | Model | Results | | | Onehot Baseline | SOTA |
|------|-------|---------|---|---|-----------------|------|
| | | **RI-T** | **Frozen** | **PT-FT** | | |
| Archive2 | RNA-FM | 0.6770 | 0.3855 | **0.8758** | 0.3481 | 0.9050 (Fu et al., 2022) |
| | RNA-MSM | 0.6610 | 0.4853 | **0.7727** | | |
| | RNA-BERT | 0.3120 | 0.3481 | **0.3351** | | |
| | SpliceBERT | 0.6126 | **0.2579** | **0.7073** | | |
| | 3UTRBERT | 0.6403 | **0.2181** | 0.5881 | | |
| | mRNA-FM | 0.5431 | 0.3693 | 0.5345 | | |
| | PlantRNA-FM | 0.5711 | 0.5491 | **0.8306** | | |
| | OmniGenome-52M | 0.5914 | 0.7046 | **0.9052** | | |
| | OmniGenome-186M | 0.5900 | 0.7284 | **0.9162** | | |
| bpRNA | RNA-FM | 0.5321 | **0.2374** | **0.7778** | 0.3024 | 0.6350(Fu et al., 2022) |
| | RNA-MSM | 0.5440 | **0.2372** | **0.6444** | | |
| | RNA-BERT | 0.4323 | 0.3405 | 0.2368 | | |
| | SpliceBERT | 0.5154 | **0.2372** | **0.6713** | | |
| | 3UTRBERT | 0.5536 | **0.2366** | 0.5283 | | |
| | mRNA-FM | 0.4425 | **0.2372** | 0.4291 | | |
| | PlantRNA-FM | 0.6441 | 0.4706 | **0.7249** | | |
| | OmniGenome-52M | 0.6421 | 0.4914 | **0.7908** | | |
| | OmniGenome-186M | 0.6968 | 0.5014 | **0.8320** | | |
| SSI | RNA-FM | 0.4405 | 0.4195 | **0.5052** | 0.1036 | 0.3720 (Gong et al., 2021) |
| | RNA-MSM | 0.4483 | 0.2700 | **0.5092** | | |
| | RNA-BERT | 0.2364 | 0.2441 | **0.3912** | | |
| | SpliceBERT | 0.2164 | 0.4244 | **0.5109** | | |
| | 3UTRBERT | 0.4719 | 0.3777 | 0.4603 | | |
| | mRNA-FM | 0.2421 | 0.3765 | 0.2397 | | |
| | PlantRNA-FM | 0.5105 | 0.4637 | **0.5286** | | |
| | OmniGenome-52M | 0.5067 | 0.4479 | **0.5621** | | |
| | OmniGenome-186M | 0.5099 | 0.4548 | **0.5711** | | |

*Table 8.* Continued from previous page (Part 2). **Black bold values** indicate underperformance relative to the `Onehot` baseline. **Orange bold values** indicate that `PT-FT` outperforms `RI-T`.

| Task | Model | Results | | | Onehot Baseline | SOTA |
|---|---|---|---|---|---|---|
| | | **RI-T** | **Frozen** | **PT-FT** | | |
| ncRNA | RNA-FM | 0.8174 | **0.6861** | **0.9634** | | |
| | RNA-MSM | 0.8027 | **0.3530** | 0.6851 | | |
| | RNA-BERT | 0.4746 | **0.1338** | **0.4940** | | |
| | SpliceBERT | 0.7242 | **0.3673** | **0.9160** | | |
| | 3UTRBERT | 0.6678 | **0.2901** | **0.7593** | 0.7807 | 0.8573 (Rossi et al., 2019) |
| | mRNA-FM | 0.8523 | **0.4415** | **0.8660** | | |
| | PlantRNA-FM | 0.7258 | **0.4438** | **0.9031** | | |
| | OmniGenome-52M | 0.8327 | **0.4600** | **0.9530** | | |
| | OmniGenome-186M | 0.8447 | **0.5188** | **0.9500** | | |
| MRL | RNA-FM | -0.2099 | **−0.0800** | **0.6010** | | |
| | RNA-MSM | -0.0923 | **−0.1190** | **0.6540** | | |
| | RNA-BERT | -0.1416 | **−0.0616** | **−0.0580** | | |
| | SpliceBERT | -0.2620 | -0.0270 | **0.6720** | | |
| | 3UTRBERT | 0.6887 | -0.0324 | **0.7068** | -0.0504 | 0.7800 (Sample et al., 2019) |
| | mRNA-FM | 0.1672 | 0.0640 | **0.2198** | | |
| | PlantRNA-FM | 0.6260 | 0.0200 | **0.7292** | | |
| | OmniGenome-52M | -0.1000 | -0.0110 | **0.7646** | | |
| | OmniGenome-186M | 0.5580 | **−0.0720** | **0.7508** | | |
| Ribozyme | RNA-FM | 0.6310 | 1.1235 | **0.4964** | | |
| | RNA-MSM | 0.6020 | 1.1371 | **0.5305** | | |
| | RNA-BERT | 0.8160 | 1.1242 | **0.7820** | | |
| | SpliceBERT | 0.8872 | 1.1177 | **0.4960** | | |
| | 3UTRBERT | 0.6133 | 1.0514 | **0.5302** | 1.1381 | - |
| | mRNA-FM | 0.5099 | 1.0711 | 0.5394 | | |
| | PlantRNA-FM | 0.5061 | 0.9231 | **0.4410** | | |
| | OmniGenome-52M | 0.6392 | 0.7360 | **0.4780** | | |
| | OmniGenome-186M | 0.5097 | 0.7166 | **0.4180** | | |

*Table 9.* Continued from previous page (Part 3). **Black bold values** indicate underperformance relative to the `Onehot` baseline. **Orange bold values** indicate that `PT-FT` outperforms `RI-T`.

| Task | Model | Results | | | Onehot Baseline | SOTA |
|------|-------|---------|---|---|-----------------|------|
| | | **RI-T** | **Frozen** | **PT-FT** | | |
| Ribozyme-ood | RNA-FM | 1.2217 | 1.2639 | **1.0163** | 1.6825 | - |
| | RNA-MSM | 1.2120 | 1.2305 | 1.2960 | | |
| | RNA-BERT | 1.2553 | **1.8507** | 1.6620 | | |
| | SpliceBERT | 1.2082 | 1.2299 | **1.0280** | | |
| | 3UTRBERT | 1.2098 | 1.1849 | **0.9277** | | |
| | mRNA-FM | 1.0518 | 1.2549 | 1.0730 | | |
| | PlantRNA-FM | 1.2230 | 1.2914 | **0.9770** | | |
| | OmniGenome-52M | 1.2554 | 1.0875 | **0.9140** | | |
| | OmniGenome-186M | 1.2210 | 0.9499 | **0.8930** | | |
| mRNA | RNA-FM | 0.0648 | 0.1065 | **0.0380** | 0.1164 | - |
| | RNA-MSM | 0.0632 | 0.1133 | **0.0538** | | |
| | RNA-BERT | 0.1168 | 0.1158 | **0.0570** | | |
| | SpliceBERT | 0.0606 | 0.0684 | **0.0370** | | |
| | 3UTRBERT | 0.0511 | 0.1074 | **0.0310** | | |
| | mRNA-FM | 0.0244 | 0.0840 | 0.0580 | | |
| | PlantRNA-FM | 0.0645 | 0.0733 | **0.0240** | | |
| | OmniGenome-52M | 0.0634 | 0.0730 | **0.0210** | | |
| | OmniGenome-186M | 0.0743 | 0.0584 | **0.0230** | | |
| mRNA-ood | RNA-FM | 0.1769 | **0.2065** | **0.0706** | 0.1594 | - |
| | RNA-MSM | 0.1760 | **0.2017** | **0.1331** | | |
| | RNA-BERT | 0.1875 | 0.1461 | **0.1490** | | |
| | SpliceBERT | 0.1755 | 0.1543 | **0.0680** | | |
| | 3UTRBERT | 0.1761 | **0.1903** | **0.0578** | | |
| | mRNA-FM | 0.0783 | **0.1873** | **0.0720** | | |
| | PlantRNA-FM | 0.1700 | **0.1937** | **0.0670** | | |
| | OmniGenome-52M | 0.1713 | **0.2156** | **0.0380** | | |
| | OmniGenome-186M | 0.1717 | **0.2130** | **0.0480** | | |

