# OpenReview forum: "Understanding Transfer Learning of RNA Foundation Models on Downstream Tasks"
_ICML.cc/2026/Conference — ICML 2026 regular_

### Official Review · Reviewer_ZBJQ · 2026-02-27

**Soundness:** 4
**Presentation:** 3
**Significance:** 4
**Originality:** 3
**Overall Recommendation:** 5
**Confidence:** 4

**Summary:**

This paper evaluates RNA sequence-processing foundation models (FM) across a wide variety of tasks, interrogating their capabilities for transfer learning either by finetuning or linear probing on final frozen embeddings in comparison to baselines (training from random initialization or trivial one-hot baseline.

Overall, the results suggest to me central dogma of molecular biology, that structure determines function, seems to be limited when it comes to RNA-FM representation learning: many models do not transfer particularly well to functional tasks and require full finetuning / overwriting of their original weights in order to be effective. Indeed biology is complex and deep learning is challenging, as evidence by how the benchmarks demonstrate a wash on performance: on some tasks the best is a finetuned model, on others finetuning dramatically deteriorates performance while frozen embeddings are the best, and in many cases the onehot baseline beats deep learning.

**Compliance With Llm Reviewing Policy:**

Affirmed.

**Final Justification:**

Thank you for the thoughtful engagement and for the interesting summary statistic results. I think it's a clear accept. Great work!

**Key Questions For Authors:**

Many readers might be more familiar with RNA when it comes to technologies like single-cell RNA-seq, and may associate "RNA" with "transcriptomics" or gene expression data. Would the authors be willing to add a section describing how the RNA sequence FMs in this work relate (or do not relate) to ML work in transcriptomics and/or how these models and/or findings connect to that field?

**Limitations:**

Yes they have. While layerwise probing could be potentially interesting, lacking it is not a deal-breaker.

**Strengths And Weaknesses:**

Strengths:
- clear and well-written, overall a thoughtful paper
- comprehensive benchmarking on a wide variety of tasks
- comprehensive benchmarking across a number of different models
- good attempt at an investigation into properties of the models / their loss landscapes / feature activations
- good use of clear baselines / sanity checks

Weaknesses:
- I think the only primary weakness of this paper is the analysis in Section 4.2, I recommend a refactor of this section.
  - First, please add a citation or two to the relevant papers around the claim that this is what happens for CNNs in computer vision so that readers can compare the methods used by CNN related work to the methods used to analyze hierarchical feature reuse here.
  - The heatmaps are cool but I don't really find heatmaps convincing, and they are hard to read and interpret (especially if colorblind).
  - As well, the math and logic is not particularly clear here, I don't exactly follow what the authors are doing as its not posited in equations.
  - Easily resolvable, so I would therefore recommend: (1) use clear mathematical notation to describe exactly what is happening in these comparisons and L2 distances between layers etc; (2) introduce reasonable summary statistics that numerically describe the properties that claimed to be seen by looking at the heatmaps (e.g. ,maybe it would be reasonable to include a "feature re-use score" versus "global adaptation score", or something to that effect); (3) keep the heatmaps for visual intuition, but also include a table with the summary statistic numbers.

Secondary concerns:
- minor: sometimes a bit too many adjectives in the writing ("crucial insight", "critical local signals")
- formatting on page 6 should have figures 2 and 3 side by side at the top for better readability

---

> ### Author Rebuttal · Authors · 2026-03-30
>
> We thank the reviewer for the concrete suggestions. We agree that Sec. 4.2 can be made clearer and more quantitative, and we will refactor it accordingly.
>
> ### **W1(a) Citations on hierarchical feature reuse in CNNs:**
> We will move the relevant CNN transfer learning citations into the main text and explicitly position our analysis relative to prior work on hierarchical feature reuse in computer vision.
>
> ### **W1(d.1 - d.3) Mathematical notation \& Summary statistics:**
> We will replace the informal description of Section 4.2 with explicit equations. Let $\theta\_l$ denote all parameters in transformer block $l$. We define normalized block drift at epoch $t$ as $d\_l^{(t)}=\frac{\|\|\theta\_l^{(t)}-\theta\_l^{(0)}\|\|\_F}{\|\|\theta\_l^{(0)}\|\|\_F}$, where $\theta\_l^{(0)}$ is the initialization (pretrained weights for PT-FT, random initialization for RI-T). At convergence ($T$), we summarize lower- and upper-layer adaptation by $D\_{\text{low}}=\frac{1}{\lfloor L/2\rfloor}\sum\_{l=1}^{\lfloor L/2\rfloor} d\_l^{(T)}$, $D\_{\text{up}}=\frac{1}{L-\lfloor L/2\rfloor}\sum\_{l=\lfloor L/2\rfloor+1}^{L} d\_l^{(T)}$, and report the hierarchical adaptation ratio $\text{HAR}=D\_{\text{low}} / D\_{\text{up}}$. $\text{HAR}<1$ indicates that lower layers move less than upper layers, while $\text{HAR}\approx 1$ indicates more global adaptation. This normalized formulation avoids confounding by raw parameter count across layers.
> | Model | Task | HAR(RI-T) | HAR(PT-FT) |
> | --- | --- | --- | --- |
> | Omni-52M | Archive2 | 1.1658 | 0.7541 |
> | Omni-186M | Archive2 | 0.9609 | 0.8012 |
> | RNA-FM | Archive2 | 1.0781 | 1.4373 |
> | RNA-BERT | Archive2 | 0.9271 | 0.6878 |
> | mRNA-FM | Archive2 | 1.0003 | 1.1594 |
>
> We will keep the heatmaps only as visual intuition and add a compact table of $D_{\text{low}}$, $D_{\text{up}}$, and $\text{HAR}$ for each representative model–task pair. We will also revise the text so that transfer-pattern descriptions are backed by both summary statistics and downstream performance, rather than by heatmap inspection alone.
>
> ### **W1(d.3) Summary statistics table alongside heatmaps:**
> We will add the tables with summary statistic numbers in the appendix in the final manuscripts.
>
> ### **Q1 (Clarification: RNA sequence FMs vs. Transcriptomics models):**
> We will add a short paragraph distinguishing RNA sequence FMs from transcriptomics/scRNA-seq models. The models studied here operate on nucleotide sequences of individual RNAs, whereas transcriptomics models often operate on gene-by-cell expression matrices. That said, the data modality, model inputs, and downstream tasks are therefore different. We will note that sequence FMs may provide complementary information for transcriptomics applications, but the findings in this paper do not directly transfer to expression-matrix modeling.
>
> ### **W2 (Minor concerns):**
> We will reduce adjective-heavy wording and place Figs. 2 and 3 side-by-side for easier comparison.

---

> > ### Author Rebuttal · Reviewer_ZBJQ · 2026-04-01
> >
> > Thank you for the thoughtful engagement and for the interesting summary statistic results. I think it's a clear accept. Great work!

---

> > > ### Author Response · Authors · 2026-04-01
> > >
> > > Thank you for the thoughtful follow-up. We are glad the additional summary-statistic analysis addressed your concerns, and we will incorporate these changes in the revision.

---

### Official Review · Reviewer_A3Bg · 2026-03-10

**Soundness:** 3
**Presentation:** 3
**Significance:** 2
**Originality:** 2
**Overall Recommendation:** 2
**Confidence:** 4

**Summary:**

This paper studies transfer learning behavior in RNA foundation models across nine downstream tasks. The authors compare four adaptation settings (Onehot, Frozen, RI-T, PT-FT) on multiple pretrained RNA transformers, and analyze transfer through frozen-feature performance, block-wise parameter changes during fine-tuning, and loss-landscape visualizations. The main conclusion is that frozen representations are not universally transferable, full fine-tuning usually works better, and pretraining appears most useful when it provides a favorable initialization for optimization, especially when pretraining and downstream tasks are well aligned.

**Compliance With Llm Reviewing Policy:**

Affirmed.

**Final Justification:**

The paper's findings are largely foreseeable from existing transfer learning literature and lack the theoretical depth expected at this venue. Categorizing observed behaviors into three patterns post-hoc does not constitute a novel contribution. The key insight — that pretraining helps more when it aligns with the downstream task — is a well-established principle, not a discovery.   This paper rediscovers the obvious.

**Key Questions For Authors:**

1. The frozen-feature analysis only uses the final hidden layer. Do the conclusions still hold under intermediate-layer probing or simple layer selection? If not, the main mechanistic claim would need to be weakened.

2. The claim that PT-FT leads to flatter minima (Finding 3) relies on qualitative loss landscape visualizations. Have the authors considered quantitative flatness measures (e.g., Hessian trace, sharpness, or SAM-based metrics)? Without these, the flatness claim is difficult to verify rigorously. If such experiments are feasible, their inclusion would substantially strengthen the paper.

3. What is the precise novel takeaway beyond the fairly standard observation that PT-FT usually outperforms Frozen and that task alignment matters? Please clarify what principle a practitioner learns here that was not already broadly expected.

4.  The "feature stagnation" pattern in RNA-BERT is attributed to pretraining–task misalignment, but RNA-BERT is also the smallest model (0.5M parameters) by a large margin. How do the authors rule out the possibility that RNA-BERT's behavior is primarily explained by insufficient capacity rather than misalignment? Is there a medium-capacity model that exhibits stagnation on an aligned task, or a small model that shows proper feature reuse? A cleaner ablation would strengthen this claim.

5.  The paper focuses exclusively on encoder-only MLM models. Given the growing adoption of autoregressive and encoder-decoder RNA models (e.g., for RNA design), do the authors expect their findings to generalize? Are there any preliminary observations that either support or challenge the generalization of the three identified transfer patterns to non-MLM pretraining objectives?

**Limitations:**

The authors discuss several limitations in the Discussion section, including the restriction to encoder-only transformers, the use of only final-layer representations for linear probing, and the open question of whether parameter-efficient fine-tuning methods could match full fine-tuning performance. These are reasonable and honest acknowledgments. However, a few additional limitations would merit explicit discussion: (1) the qualitative nature of the loss landscape analysis and its sensitivity to the random projection directions; (2) the limited diversity of OOD evaluation settings; and (3) the potential confound between model capacity and transfer pattern classification (particularly for RNA-BERT). The paper does not raise any significant ethical concerns.

**Strengths And Weaknesses:**

# Strengths

- **Soundness**. The paper is empirically solid overall. The experiments are organized around clear hypotheses, the evaluation covers a relatively broad set of RNA tasks, and the comparison across Onehot, Frozen, RI-T, and PT-FT is useful. The study is careful enough to support the basic empirical takeaway that full fine-tuning is generally more reliable than treating pretrained RNA models as frozen feature extractors.

- **Presentation**. The paper is reasonably well structured and easy to follow. The hypothesis-driven organization helps, and the main findings are clearly stated. The experimental protocol is mostly understandable, and the figures support the overall narrative.

-  **Significance**. Understanding when RNA pretraining helps is important for the RNA FM community, especially given the rapid growth of pretrained sequence models.

# Weaknesses

- **Originality**. My main concern is that the central findings are fairly unsurprising. The paper ultimately shows that frozen final-layer features are often insufficient, full fine-tuning usually performs better, and pretraining helps more when it is aligned with the downstream task. These are broadly consistent with common experience in transfer learning, and the paper does not extract a substantially deeper or more actionable principle beyond this. As a result, the work feels more like a careful confirmation study than a genuinely new mechanistic insight.

- **Soundness**. Some of the stronger claims are not fully justified by the evidence provided. The conclusion that pretraining mainly helps through optimization rather than feature reuse is plausible, but the paper only probes the final hidden layer and relies on parameter-distance and qualitative loss-landscape analyses. This is not enough to rule out alternative explanations, such as useful features residing in intermediate layers, limitations of the probing setup, or architecture/head mismatch. The loss-landscape evidence is also qualitative and should not carry too much mechanistic weight.

- **Scope**. The study is limited to a fairly specific class of encoder-style RNA transformers and mostly last-layer probing. That makes the title and some conclusions feel broader than the actual evidence supports.

---

> ### Author Rebuttal · Authors · 2026-03-30
>
> We thank the reviewer for the constructive feedback.
>
> ### **Q1 (Intermediate-layer probing analysis):**
> We performed additional coarse frozen probing at four representative depths {L/4, L/2, 3L/4, Final} on Archive2 across five models (see the result Table in `Gk8u` **W3**). These results refine rather than overturn our original conclusion: intermediate layers can outperform the final layer, confirming that last-layer-only probing can understate frozen-feature utility. However, PT-FT still exceeds the best of these four probed depths for 4/5 models and is tied only for RNA-BERT. We will therefore narrow the claim from “frozen representations are insufficient” to the more precise statement that final-layer frozen representations are not a reliable default, and that simple depth selection over these four depths does not overturn the empirical advantage of PT-FT on this tested task. More expressive multi-layer fusion or task-specific heads may further improve frozen baselines.
>
> ### **Q2 (Quantitative flatness):**
> We agree that the original loss landscape is qualitative and should not carry the full weight of the flatness claim. We thus evaluated sharpness and SAM-based metrics on the Archive2 task. PT-FT shows substantially lower sharpness than RI-T in the clearest aligned cases (Omni-52M: 8.61 vs 26.32; Omni-186M: 3.84 vs 6.28; RNA-FM: 7.30 vs 20.77); while evidence for weaker/misaligned model-task pairs is more mixed. We will therefore soften *Finding 3* to be task-conditional: in many tested settings, especially structure-aligned ones, pretraining behaves primarily as an optimization prior. The loss landscape plots will remain as qualitative intuition only.
>
> | Model | Task | Setting | Base Loss ↓ | Sharpness ↓ | SAM Metric ↓ |
> | --- | --- | --- | --- | --- | --- |
> | Omni-52M | Archive2 | PT-FT | 0.5764 | 8.6111 | 0.0011 |
> | Omni-52M | Archive2 | RI-T | 0.5832 | 26.3233 | 0.0039 |
> | Omni-186M | Archive2 | PT-FT | 0.5871 | 3.8365 | 0.0023 |
> | Omni-186M | Archive2 | RI-T | 0.6801 | 6.2805 | 0.0043 |
> | RNA-FM | Archive2 | PT-FT | 0.5799 | 7.3025 | 0.0012 |
> | RNA-FM | Archive2 | RI-T | 0.9240 | 20.7655 | 0.0034 |
> | mRNA-FM | Archive2 | PT-FT | 0.7934 | 22.4817 | 0.0214 |
> | mRNA-FM | Archive2 | RI-T | 0.7951 | 30.6932 | 0.0288 |
>
> ### **Q3/W1 (Originality & Takeaway):**
> We agree that isolated observations like “PT-FT often outperforms Frozen” are not, by themselves, surprising. The intended novelty is instead the systematic decomposition across onehot/frozen/RI-T/PT-FT over a broad RNA-task benchmark, which reveals three practically distinct transfer regimes: aligned reuse, global adaptation, and adverse/confounded stagnation. The actionable takeaway for practitioners is therefore not simply “fine-tune more”, but that adaptation strategy should depend on pretraining–task alignment: structure-aligned settings can benefit from reusable representations, whereas weakly aligned functional/OOD settings typically require end-to-end adaptation, and apparent shallow-layer stability should not be interpreted as productive reuse. We will revise the paper to foreground this narrower, more actionable contribution.
>
> ### **Q4 (Confounded adverse-transfer for RNA-BERT):**
> We agree that RNA-BERT alone does not cleanly disentangle task misalignment from severe capacity limitation, esp. given its much smaller parameters count. We will therefore weaken the original wording and no longer present RNA-BERT as stand-alone evidence for misalignment-driven stagnation. We will describe RNA-BERT as a confounded adverse-transfer case where low capacity likely contributes. Our broader takeaway is limited to the observed pattern that transfer behavior depends on alignment and capacity, not to a clean causal attribution for RNA-BERT. Larger models on weakly aligned tasks still show that mismatch can induce global overwriting rather than productive reuse, but this is distinct from the specific causal interpretation of RNA-BERT.
>
> ### **Q5/W3 (Scope):**
> Our evidence currently supports conclusions for the evaluated family of encoder-only RNA sequence transformers. We do not claim direct generalization to autoregressive or encoder-decoder RNA FMs, and we will make this limitation explicit in the title/discussion. We will also add results on ERNIE-RNA and RiNALMo as supporting evidence within the encoder-style family rather than proof about non-MLM or generative RNA FMs.
>
> ### **Limitations:**
> We thank the reviewer for the suggested limitations and will add them explicitly in the revision: (i) 2D loss-landscape visualizations depend on projection directions and remain qualitative; (ii) our OOD coverage is limited mainly to mutation-based shifts and does not span the full diversity of biologically relevant distribution shifts; and (iii) transfer-pattern interpretation remains partially confounded with model capacity, especially for RNA-BERT. The new quantitative flatness metrics address only the first point partially and do not remove the latter two.

---

> > ### Author Rebuttal · Reviewer_A3Bg · 2026-04-07
> >
> > Thanks for the rebuttal.
> >
> > The paper's findings are largely foreseeable from existing transfer learning literature and lack the theoretical depth expected at this venue. Categorizing observed behaviors into three patterns post-hoc does not constitute a novel contribution. The key insight — that pretraining helps more when it aligns with the downstream task — is a well-established principle, not a discovery. I vote for rejection.

---

> > > ### Author Response · Authors · 2026-04-07
> > >
> > > Thank you for the follow-up. We understand that the remaining disagreement is now primarily about the kind and threshold of originality that should be credited to this work, rather than about unresolved experimental soundness. As reflected in the public discussion, we narrowed and clarified the framing of the paper from a broad mechanistic account to a systematic empirical characterization of transfer behavior in current encoder-only RNA sequence transformers, and responded to the main soundness and scope concerns through the additional analyses and clarifications already in the discussion record. We therefore hope the contribution can be assessed in this more precise empirical-analysis framing.

---

### Official Review · Reviewer_THX7 · 2026-03-10

**Soundness:** 3
**Presentation:** 3
**Significance:** 3
**Originality:** 3
**Overall Recommendation:** 4
**Confidence:** 4

**Summary:**

The paper systematically investigates the mechanisms underlying transfer learning in RNA foundation models, addressing a critical gap in understanding why and how pretraining benefits downstream structural and functional tasks. While large-scale pretraining has become a dominant paradigm in sequence biology, the authors challenge the prevailing assumption that these models function as universal, off-the-shelf feature extractors. To dissect this, the study evaluates multiple transformer-based RNA models across diverse downstream tasks utilizing a standardized benchmark platform. The methodology contrasts different adaptation strategies while employing advanced analytical techniques like block-wise parameter distance tracking and loss landscape geometry visualization to capture the full temporal dynamics of adaptation.

The primary contribution of this work is the empirical demonstration that the traditional paradigm of hierarchical feature reuse, which is widely accepted in computer vision, does not universally apply to RNA foundation models. The authors reveal that the utility of frozen pretrained representations is strictly task-dependent; while they perform adequately on structure-aligned tasks, they frequently underperform simple one-hot encodings on functional tasks that rely on local sequence statistics.

The authors redefine the core utility of current RNA pretraining, concluding that it primarily serves as a highly effective optimization prior rather than a robust source of static, reusable features. The study shows that when task alignment and model capacity are sufficient, starting from pretrained weights places the network in a highly favorable region of the parameter space.

**Compliance With Llm Reviewing Policy:**

Affirmed.

**Final Justification:**

My concerns have been adequately addressed.

**Key Questions For Authors:**

Yet, the study evaluates only standard MLM-trained models and omits highly relevant, concurrent models explicitly designed with structural priors or multi-modal objectives (e.g., ERNIE-RNA, which is cited, or RiNALMo). How does the exclusion of these explicitly structure-infused models affect the generalizability of your claim that the CV paradigm of feature reuse does not generally extend to RNA FMs?

**Limitations:**

Yes

**Strengths And Weaknesses:**

Strengths

1. The submission evaluates models across a highly diverse set of 9 downstream tasks. This includes both structure prediction and function predictio, incorporating both token-level and sequence-level objectives, as well as in-distribution and out-of-distribution splits.
2. Instead of merely comparing initial and final model weights, the authors track the block-wise l2 distance dynamically across every training epoch. Furthermore, coupling this with loss landscape geometry visualizations provides empirical, geometric proof of their optimization claims.
3. This paper is clearly written and highly motivated.

Weakness

1. The study explicitly restricts its analysis to small-to-medium-sized, encoder-only Transformer architectures. By excluding larger autoregressive models or models with explicit structural priors integrated into the architecture, the universality of the claim regarding capacity-induced optimization instability is somewhat constrained.

2. Why not include RiNALMo and ERNIE-RNA into the experiments?

---

> ### Author Rebuttal · Authors · 2026-03-30
>
> We thank the reviewer for highlighting the scope issue. We agree that our original wording was broader than the evidence supports, and in the revision we will explicitly scope the paper to the evaluated family of encoder-only RNA sequence transformers rather than all RNA FMs.
>
> ### **Q1/W2 (Additional experiments on RiNALMo and ERNIE-RNA models):**
> Following the reviewer's suggestion, we additionally evaluated RiNALMo and ERNIE-RNA on 5 tasks. And we also performed additional coarse probing at four representative depths {L/4, L/2, 3L/4, Final} on the Archive2 task. These results refine rather than overturn our conclusion: **explicit structural priors strengthen transfer on structure-aligned tasks and can make frozen layer representations more competitive**, but they still do not make CV-style shallow-feature reuse a reliable default across tasks. We will therefore revise our claim to the narrower statement that, within current encoder-style RNA sequence FMs, hierarchical feature reuse is not a universal or dependable assumption; transfer remains strongly dependent on pretraining–task alignment and task type.
>
> **We would also like to clarify that the original benchmark was not restricted to plain-MLM models:** it already included models whose pretraining incorporates RNA secondary-structure alignment signals, such as PlantRNA-FM and the OmniGenome series, although we agree that architecture-level structure-infused models were underrepresented in the original submission. We will make this distinction explicit.
> | Model | Setting | Archive2 ↑ | bpRNA ↑ | SSI ↑ | ncRNA ↑ | MRL ↑ |
> | --- | --- | --- | --- | --- | --- | --- |
> |  | one-hot | 0.3481 | 0.3024 | 0.1036 | 0.7807 | -0.0504 |
> | RiNALMo | Frozen | 0.7248 | 0.6298 | 0.0258 | 0.8435 |  -0.0224 |
> | RiNALMo | RI-T | 0.6649 | 0.2366 | -0.007 | 0.075 | -0.2021 |
> | RiNALMo | PT-FT | 0.9434 | 0.8497 | 0.2978 | 0.9565 | 0.7818 |
> | ERNIE-RNA | Frozen | 0.3680 | 0.3302 | 0.0085 | 0.5004 | -0.0878 |
> | ERNIE-RNA | RI-T | 0.7351 | 0.5860 | -0.0043 | 0.3881 | -0.1265 |
> | ERNIE-RNA | PT-FT | 0.9223 | 0.8572 | 0.2194 | 0.9504 | 0.6550 |
>
> | Model | Task | L/4 | L/2 | 3L/4 | Final Layer |
> | --- | --- | --- | --- | --- | --- |
> | RiNALMo | Archive2 | 0.565 $\pm$ 0.005 | 0.610 $\pm$ 0.001 | 0.695 $\pm$ 0.003 | 0.723 $\pm$ 0.006 |
> | ERNIE-RNA | Archive2 | 0.484 $\pm$ 0.015 | 0.588 $\pm$ 0.011 | 0.607 $\pm$ 0.008 | 0.384 $\pm$ 0.016 |
>
> ### **W1 (The universality of the claim):**
> We do not claim direct generalization to autoregressive or encoder–decoder RNA models, whose objectives and evaluation protocols differ substantially from the discriminative encoder setting studied here. We will state this limitation explicitly and present any new ERNIE-RNA/RiNALMo results only as additional evidence within the encoder-style family, not as proof about all RNA pretraining objectives.

---

> > ### Author Rebuttal · Reviewer_THX7 · 2026-04-03
> >
> > Thank you for your response. I keep my positive score.

---

> > > ### Author Response · Authors · 2026-04-06
> > >
> > > We sincerely thank you for your acknowledgment and for maintaining a positive score. We greatly appreciate your time and constructive feedback, which have helped us improve our work.

---

### Official Review · Reviewer_Gk8u · 2026-03-13

**Soundness:** 3
**Presentation:** 3
**Significance:** 3
**Originality:** 3
**Overall Recommendation:** 4
**Confidence:** 4

**Summary:**

This paper does not propose a new RNA foundation model, but systematically studies the transfer learning mechanism of existing RNA foundation models. On several structural and functional tasks, the author makes a unified evaluation of various existing encoder-only RNA Transformer, and compares the four settings of Onehot, Frozen, RI-T and PT-FT. The main conclusions of this paper are as follows:

1) The validity of freezing representation is highly dependent on the task;
2) Hierarchical feature reuse in similar vision is not universally established in RNA model;
3) The benefits of pre-training are often more like providing better optimization initialization than directly providing a general representation of freezable reuse. On the whole, this is a mechanism analysis /benchmark paper.

**Compliance With Llm Reviewing Policy:**

Affirmed.

**Final Justification:**

The additional analyses are helpful and address several of my concerns.
But the main issues remain: the new evidence is still limited to a few representative settings, and the central claims are better supported as empirical observations than as general mechanisms.

**Key Questions For Authors:**

1) Did you run any basic per-model hyperparameter sensitivity analysis to ensure that the main conclusions are not an artifact of using a single unified tuning protocol across all models?

2) For the central claim that pretraining mainly acts as an optimization prior, can you provide more quantitative evidence, such as sharpness or curvature metrics, beyond loss landscape visualizations?

**Limitations:**

yes

**Strengths And Weaknesses:**

Strengths

1) RNA foundation models are widely used, but there is a lack of systematic research on why pre-training is effective. This paper directly discusses this core issue, which has practical significance.

2) The four settings (Onehot/Frozen/RI-T/PT-FT) can clearly distinguish the architectural ability, pre-training benefit, frozen characterization ability and fine-tuning benefit. The task covers structure, function and OOD generalization at the same time, and the design matches the problem of the paper.

3) The author uses block-wise ℓ2 distance and loss landscape analysis to discuss feature reuse and initialization effect, which makes the work go beyond simple performance comparison.

4)  The corresponding relationship among question raising, experiment setting, result analysis and discussion is clear, and the overall narrative is smooth.

Weaknesses

1) The current evidence is more like empirical support than strong mechanism proof. In particular, the loss landscape analysis is biased, which can support the directional conclusion, but not enough to exclude other explanations.

2) The difference between different models is not only whether the pre-training is "task-aligned", but also mixed with factors such as parameter quantity, tokenization, pre-training data, species range and training objectives. It is more like systematic correlation analysis than causal analysis under strictly controlled variables.

3) At present, only the last layer of hidden state is used for probing. If the middle layer or multi-layer fusion is more suitable for downstream tasks, the conclusion that "frozen representations are insufficient" needs to be more cautious. The author admits this in limitations.

4) Unified protocol is helpful for comparability, but models of different sizes and sources may not share the same set of optimal hyperparameters. The "most experiments" in the appendix, instead of using multiple random seeds for all experiments, also slightly weakened the robustness.

5) It is mainly the level of empirical analysis, not the breakthrough of methods.

---

> ### Author Rebuttal · Authors · 2026-03-30
>
> We thank the reviewer for the careful and constructive feedback. We agree that the current paper should be framed as a systematic empirical characterization of transfer behavior in current encoder-only RNA sequence transformers, rather than as a strictly controlled causal proof, and we will revise the wording throughout accordingly.
>
> ### **Q1/W4 (Hyperparameter sensitivity):**
> Our unified protocol was chosen to maximize cross-model comparability, not to claim per-model optimal tuning. To test whether the key PT-FT vs. RI-T conclusion is an artifact of a single default learning-rate (LR) choice, we ran a targeted LR sweep on the representative structure-aligned task **Archive2** for OmniGenome-52M, RNA-FM, and mRNA-FM, using PT-FT LRs [1e−5,2e−5,5e−5] and RI-T LRs [1e−5,1e−4,1e−3], averaged over 3 seeds.
> | Model | Training |1e-5 | 2e-5 | 5e-5 |
> | --- | --- | --- | --- | --- |
> | Omni-52M | PT-FT | 0.9072 | 0.9134 | 0.9250 |
> | RNA-FM | PT-FT | 0.8245 | 0.8509 | 0.8775 |
> | mRNA-FM | PT-FT | 0.6878 | 0.7086 | 0.7174 |
>
> | Model | Training | 1e-3 | 1e-4 | 1e-5 |
> | --- | --- | --- | --- | --- |
> | Omni-52M | RI-T | 0.7505 | 0.7918 | 0.6453 |
> | RNA-FM | RI-T | 0.5987 | 0.7195 | 0.6799 |
> | mRNA-FM | RI-T | 0.3546 | 0.4527 | 0.6585 |
>
> Two points emerge.
> - PT-FT remains stronger than RI-T across all tested settings: for each of the three models, even the worst PT-FT LR outperforms the best RI-T LR.
> - RI-T is substantially more sensitive to learning-rate choice than PT-FT, so per-model tuning can change some absolute gaps and, in some cases, the ordering between RI-T and Frozen, but it does not remove the advantage of starting from pretrained weights on this representative task.
>
> We will revise the text to make the narrower claim that our conclusions are relative under a unified evaluation protocol and are most robust for the PT-FT vs. RI-T comparison, rather than claiming that one default hyperparameter setting is optimal for every model.
>
> ### **Q2/W1 (Quantitative evidence for optimization prior hypothesis):**
> We agree that the 2D loss-landscape plots are qualitative only. We therefore added quantitative flatness analyses using sharpness and a SAM-style perturbation metric (see result Table in `A3Bg` **Q2**). On Archive2, PT-FT shows substantially lower sharpness than RI-T in the clearest aligned cases (Omni-52M: 8.61 vs 26.32; Omni-186M: 3.84 vs 6.28), while other tested model pairs are weaker/mixed.
>
> We will therefore revise *Finding 3* to the narrower statement that, in many tested settings (especially structure-aligned ones) pretraining behaves primarily as an optimization prior, rather than presenting this as a universal mechanism across all tasks. The landscape plots will remain as qualitative illustrations, not stand-alone proof.
>
> ### **W2 (Causal ablation):**
> We agree that the present study is a systematic correlational analysis across public RNA FMs rather than a single-factor causal ablation: model capacity, tokenization, pretraining data, and objectives vary jointly. We will make this scope explicit and replace causal wording such as "prove/demonstrate" with "provide evidence consistent with".
>
> ### **W3 (Intermediate-layer probing analysis):**
> We performed additional coarse frozen probing at four representative depths {L/4, L/2, 3L/4, Final} on Archive2 across five models. These results refine rather than overturn our original conclusion: intermediate layers can outperform the final layer, confirming that last-layer-only probing can understate frozen-feature utility. However, PT-FT still exceeds the best of these four probed depths for 4/5 models and is essentially tied only for RNA-BERT. We will therefore narrow the claim from “frozen representations are insufficient” to the more precise statement that final-layer frozen representations are not a reliable default, and that simple depth selection over a small set of representative layers does not overturn the empirical advantage of PT-FT on this representative task.
> | Model | Task | L/4 | L/2 | 3L/4 | Final Layer |
> | --- | --- | --- | --- | --- | --- |
> | Omni-52M | Archive2 | 0.440 $\pm$ 0.018 | 0.577 $\pm$ 0.010 | 0.646 $\pm$ 0.008 | 0.721 $\pm$ 0.001 |
> | Omni-186M | Archive2 | 0.547 $\pm$ 0.005 | 0.614 $\pm$ 0.006 | 0.669 $\pm$ 0.005 | 0.726 $\pm$ 0.001 |
> | RNA-BERT | Archive2 | 0.296 $\pm$ 0.038 | 0.299 $\pm$ 0.022 | 0.293 $\pm$ 0.026 | 0.335 $\pm$ 0.039 |
> | RNA-FM | Archive2 | 0.602 $\pm$ 0.010 | 0.626 $\pm$ 0.011 | 0.617 $\pm$ 0.012 | 0.377 $\pm$ 0.031 |
> | mRNA-FM | Archive2 | 0.473 $\pm$ 0.005 | 0.476 $\pm$ 0.004 | 0.476 $\pm$ 0.004 | 0.350 $\pm$ 0.001 |
>
> ### **W5 (Position of the paper):**
> We agree that the contribution is empirical analysis rather than a new learning method. We will revise the framing accordingly. The intended contribution is the systematic decomposition across one-hot/frozen/RI-T/PT-FT and the resulting task-dependent transfer regimes, which we believe is practically useful for the RNA FM community.

---

> > ### Author Rebuttal · Reviewer_Gk8u · 2026-04-02
> >
> > Thank you for the detailed rebuttal. The additional analyses are helpful and address several of my concerns.
> >
> > I will keep my current score, however, since the main issues remain: the new evidence is still limited to a few representative settings, and the central claims are better supported as empirical observations than as general mechanisms.

---

> > > ### Author Response · Authors · 2026-04-06
> > >
> > > Thank you again for the thoughtful follow-up and for clarifying your current assessment. To address your remaining point about the limited range of representative settings, we extended the sanity checks beyond `Archive2` to one functional in-distribution task (`ncRNA`) and one OOD task (`mRNA-ood`). The same broad empirical pattern persists (see result tables [here](https://zenodo.org/records/19437942)):
> > >
> > > - Under per-model LR sweeps, best PT-FT remains stronger than best RI-T in all 10 added tasks across the tested models (for `mRNA-ood`, lower is better).
> > > - Quantitative flatness also mostly favors PT-FT on the added tasks, with lower SAM-style metrics in all 10 added pairs and lower sharpness in 9/10, the only mixed case being RNA-BERT on `ncRNA`.
> > > - Coarse frozen probing at {L/4, L/2, 3L/4, Final} again shows that intermediate layers can improve over the final layer, but the best frozen depth still does not overturn the PT-FT advantage in any of the 10 added settings.
> > >
> > > We agree that these extensions strengthen the empirical pattern rather than establish a general mechanism, and we will keep that narrower framing explicit in the revision.

---

### Decision · Program_Chairs · 2026-04-30

**Decision:**

Accept (regular)

**Comment:**

This paper examines why transfer learning is effective in RNA biology.

Strengths

- RNA foundation models are widely used, but there is a lack of systematic research on why pre-training is effective. This paper directly discusses this core issue, which has practical significance. Understanding when RNA pretraining helps is important for the RNA FM community, especially given the rapid growth of pretrained sequence models.
- The four settings (Onehot/Frozen/RI-T/PT-FT) can clearly distinguish the architectural ability, pre-training benefit, frozen characterization ability and fine-tuning benefit. The task covers structure, function and OOD generalization at the same time, and the design matches the problem of the paper.
- The blockwise L2 distance analysis clearly demonstrates that some but not all tasks benefit from hierarchical feature transfer

Weaknesses

- The submission only evaluates relatively-small, encoder-only models
- Some reviewers comment that the results are confirmatory rather than surprising. While this may be true for broad conclusions such as finetuning almost always providing stronger results, it is nevertheless valuable to have this confirmed and documented. In addition, the paper does much deeper analysis about the hierarchical feature transfer hypothesis, and the rebuttal shows evidence for interesting layer-wise behavior as well.

Overall, this is a well-executed paper that examines the assumptions in an important subfield of applied ML and provides both practical and mechanistic insights.